# The Velocity Deficit: Initial Energy Injection for Flow Matching

**Linze Li** [1]  **Zong-Wei Hong** [1]  **Shen Zhang** [1]  **Bo Lin** [1]  **Jinglun Li** [1]  **Yao Tang** [†1]  **Jiajun Liang** [1]

## Abstract

While Flow Matching theoretically guarantees constant-velocity trajectories, we identify a critical breakdown in high-dimensional practice: the Velocity Deficit. We show that the MSE objective systematically underestimates velocity magnitude, causing generated samples to fail to reach the data manifold—a phenomenon we term Integration Lag. To rectify this, we propose Initial Energy Injection, instantiated via two complementary methods: the training-based Magnitude-Aware Flow Matching (MAFM) and the training-free Scale Schedule Corrector (SSC). Both are grounded in our discovery of a crucial asymmetry: velocity contraction causes harmful kinetic stagnation at the trajectory's start, yet acts as a beneficial denoising mechanism at its end. Empirically, SSC yields significant efficiency gains with zero retraining and just one line of code. On ImageNet-1k ($256 \times 256$), it improves FID by 44.6% (from 13.68 to 7.58) and achieves a $5\times$ speedup, enabling a 50-step generator (FID 7.58) to beat a 250-step baseline (FID 8.65). Furthermore, our methods generalize to Text-to-Image tasks and high-resolution generation, improving FID on MS-COCO by $\sim$22%.

## 1. Introduction

Flow Matching (FM) ([Lipman et al., 2023](#)) has rapidly emerged as a powerful paradigm for training Continuous Normalizing Flows (CNFs), offering a robust alternative to diffusion models. By regressing a time-dependent vector field $v(x, t)$, FM constructs a probability path that transports a simple source distribution $p_0$ (noise) to a complex target distribution $p_1$ (data). Theoretically, Flow Matching constructs deterministic straight-line paths, implying that particles should traverse from $t = 0$(noise) to $t = 1$(data) with a constant velocity vector $v_{target} = x_1 - x_0$. Consequently, the optimal velocity magnitude should be time-invariant: $\|v_{target}\| \equiv \|x_1 - x_0\|$.

**Velocity Deficit.** Despite the theoretical guarantee of straight-line paths, we identify a systematic discrepancy between the ideal transport dynamics and those learned by neural networks in practice. While prior works have focused on rectifying the *curvature* of trajectories (e.g., via Reflow([Liu et al., 2023](#)) or distillation([Salimans & Ho, 2022](#))), we reveal a more fundamental dynamical bias: the Velocity Deficit. As illustrated in Figure 1(a), we observe that in high-dimensional spaces, FM models consistently *underestimate* the required velocity magnitude. Instead of maintaining the constant target norm $\|x_1 - x_0\|$, the predicted magnitude $\|v_\theta(x, t)\|$ exhibits a monotonic decay, dropping from the noise norm $\|x_0\|$ to the data norm $\|x_1\|$. Crucially, under the assumption of high-dimensional orthogonality([Ledoux, 2001](#)), both norms are strictly smaller than the target transport distance ($\sqrt{\|x_1\|^2 + \|x_0\|^2}$). In [section 3](#), we formally analyze this phenomenon, attributing it to the properties of the Mean Squared Error (MSE) objective in high-dimensional regression.

**Consequence: Integration Lag.** This systematic underestimation of velocity has severe consequences for generation quality. During numerical integration (e.g., Euler sampling), the deficient velocity field fails to push particles effectively across the probability gap within the unit time interval. We term this phenomenon Integration Lag. As a result, generated samples often fail to fully converge to the data manifold, remaining "upstream" in the flow. Qualitatively, this manifests as residual artifacts, incomplete objects or distorted local details, as shown in Figure 1(b).

A naive solution to mitigate this lag is to explicitly rectify the predicted velocity magnitude. However, we empirically find that a global, uniform boost is suboptimal as it tends to re-inject noise and over-smooth high-frequency details during the final stages of generation. This suggests that the impact of the velocity deficit is not uniform but evolves dynamically across the trajectory. This leads to our key observation: **Asymmetric Energy Injection.** While the MSE objective drives velocity magnitude decay globally, our analysis reveals that this contraction plays fundamentally opposing roles at the trajectory boundaries:

---

† Project Leader. [1]Jiiov Technology, Beijing, China. Correspondence to: Linze Li <llzinzju@gmail.com>.

*Proceedings of the 43rd International Conference on Machine Learning*, Seoul, South Korea. PMLR 306, 2026. Copyright 2026 by the author(s).

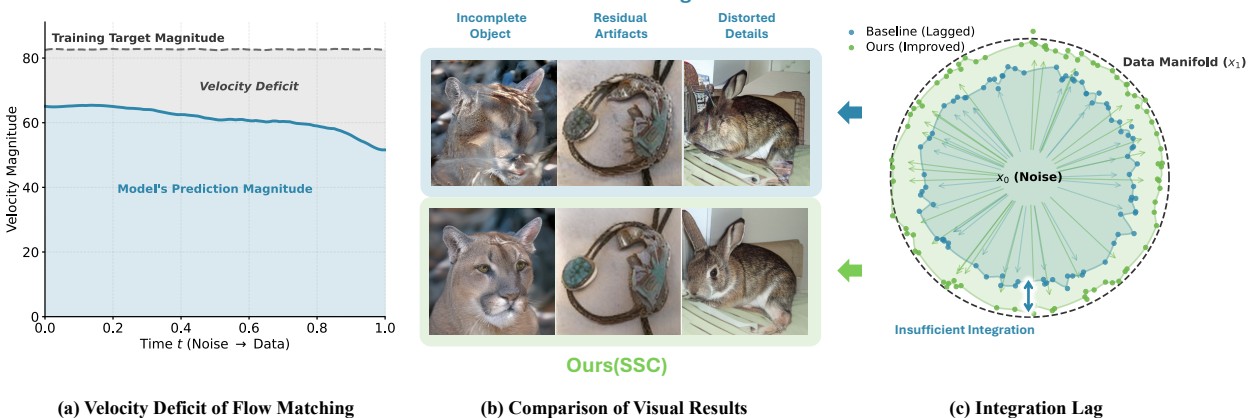

(a) Velocity Deficit of Flow Matching    (b) Comparison of Visual Results    (c) Integration Lag

*Figure 1.* **The Velocity Deficit Phenomenon and Correction Strategy. (a)** On ImageNet-1k $256 \times 256$, the learned baseline magnitude (Blue) of SiT decays monotonically, deviating from the constant OT target and creating a **Velocity Deficit** (Shaded Area), which represents the missing kinetic energy required for transport. **(b)** The velocity deficit leads to pronounced visual degradation in baseline samples, including **Residual Artifacts**, **Incomplete Object**, and **Distorted Details**. SSC effectively mitigates these failures by initial energy injection. **(c)** Standard flow matching models (Blue) fail to generate trajectories that reach the target data manifold (Black Ring), resulting in a significant **Integration Lag**. Our correction (Green) compensates for the signal starvation, drastically reducing this lag.

- **At $t \to 0$ (Noise end), the contraction is Harmful (Signal Starvation):** The model underestimates the signal component, preventing particles from initiating transport. This leads to structurally deficient samples (high FID) as the solver fails to reach the data manifold.

- **At $t \to 1$ (Data end), the contraction is Beneficial (Detail Preservation):** Here, the velocity decay acts as an implicit denoising mechanism. While boosting the scale at the end can reduce noise, it risks over-smoothing high-frequency details. Therefore, respecting the natural contraction is crucial for preserving sharp textures and ensuring realistic convergence.

**Proposed Solution.** Based on this insight, we argue that the solution lies in Asymmetric Energy Injection rather than global magnitude forcing. In this paper, we introduce two complementary solutions: (1) **Magnitude-Aware Flow Matching (MAFM)**: A training-based objective that augments the standard loss with explicit magnitude supervision, forcing the model to overcome magnitude collapse during training. (2) **Scale Schedule Corrector (SSC)**: A training-free, 'plug-and-play' inference intervention that applies a time-dependent scaling factor $\gamma(t)$ to the velocity field. SSC injects kinetic energy at the beginning ($t \to 0$) to compensate for integration lag while annealing to identity at the end ($t \to 1$).

**Contributions.** Our main contributions are as follows:

1. **Theoretical Analysis:** We formally identify and define the *Velocity Deficit*, characterizing it as a structural bias of MSE minimization that leads to Integration Lag in high dimensions. Crucially, we uncover a dynamical

asymmetry where velocity contraction is harmful at the start but beneficial for denoising at the end.

2. **Methodology:** Based on the discovery of asymmetry, we propose the principle of **Initial Energy Injection**. We provide two instantiations: **Magnitude-Aware Flow Matching (MAFM)** for training and the **Scale Schedule Corrector (SSC)** for inference. SSC serves as a plug-and-play method that requires zero retraining and negligible overhead.

3. **Efficiency & Performance:** We demonstrate that SSC significantly boosts performance in efficiency-critical regimes. On ImageNet-1k $256 \times 256$, it reduces the FID from 13.68 to 7.58 at NFE(Number of Function Evaluations)=50, a 44.6% improvement. This outperforms the 250-step baseline (FID 8.65), representing a $5\times$ **speedup**. Furthermore, our methods generalize to Text-to-Image tasks and high-resolution generation, improving MS-COCO FID by $\sim$22% (FID from 6.03 to 4.71).

## 2. Related Work

### 2.1. Flow Matching and Continuous Generative Models

Generative modeling has evolved from discrete-time frameworks (Song & Ermon, 2019; Ho et al., 2020) to continuous formulations grounded in Neural ODEs (Chen et al., 2018). While (Song et al., 2021) unified these through Stochastic Differential Equations (SDEs), Flow Matching (FM) (Lipman et al., 2023) and concurrent Stochastic Interpolants (Albergo & Vanden-Eijnden, 2023) offer a simulation-free alternative by regressing a time-dependent vector field $v_t(x)$.

This vector field drives the ODE(Ordinary Differential Equation) solver to transport a simple source distribution $p_0$ to the complex data distribution $p_1$. Theoretically, FM aims to recover Optimal Transport (OT) displacement interpolants (Benamou & Brenier, 2000), implying straight trajectories with constant velocity. While recent scalable architectures (Peebles & Xie, 2023; Ma et al., 2024) demonstrate impressive synthesis results, they often overlook a critical discrepancy: the learned vector field rarely matches the ideal OT paths in practice. Our work identifies a systematic Velocity Deficit, where the trained vector field fails to maintain sufficient kinetic energy to traverse the manifold, necessitating a rethinking of the flow dynamics.

## 2.2. Trajectory Straightening and Velocity Attenuation

To accelerate sampling, extensive research focuses on "straightening" the probability paths. EDM (Karras et al., 2022) highlighted that the curvature of ODE trajectories significantly amplifies truncation errors, necessitating carefully designed noise schedules. Building on this, Rectified Flow (ReFlow) (Liu et al., 2023) and its variants (Liu et al., 2024) employ a recursive "ReFlow" procedure to distill curved ODE trajectories into straighter paths. Similarly, Consistency Models (Song et al., 2023) aim to map noise to data in a single step. While these methods effectively reduce geometric curvature, they neglect the *magnitude deficit* arising from the inherent uncertainty of coupling in high-dimensional spaces. Since the neural network learns the conditional expectation of target velocities, the Mean Squared Error (MSE) objective inevitably averages out conflicting crossing trajectories, leading to a contractive bias in the vector field's magnitude. Unlike ReFlow which requires expensive retraining to fix curvature, our proposed Scale Schedule Corrector (SSC) directly addresses this *integration lag* by compensating for the velocity attenuation, ensuring particles reach the target distribution.

## 2.3. Weighting Strategies and Training Dynamics

The weighting schedule $\lambda(t)$ in the training objective $\mathbb{E}_t[\lambda(t)\mathcal{L}_t]$ is pivotal for model convergence. Previous works in diffusion models, such as Min-SNR (Hang et al., 2023), address the conflicting gradients across timesteps to balance optimization difficulty. Similarly, Logit-Normal sampling (Esser et al., 2024) concentrates training capacity on perceptually critical intervals. However, these strategies function primarily as variance reduction or optimization balancing schemes; they do not enforce physical constraints on the flow's energy. In contrast, our Magnitude-Aware Flow Matching(MAFM) imposes a strong physical inductive bias: it explicitly penalizes magnitude collapse in the signal-starved early stage. Instead of merely balancing gradient variance, MAFM forces the model to recover the "missing momentum" required to initiate transport.

## 2.4. Inference-Time Interventions

Guidance techniques are standard for enhancing generation fidelity at sampling time. Classifier-Free Guidance (CFG) (Ho & Salimans, 2022) extrapolates between conditional and unconditional estimates, implicitly boosting the velocity magnitude to align samples with high-density regions. However, excessive guidance often causes over-saturation. Methods like CFG-Rescale explicitly reduce the velocity norm to match pixel-space standard deviation, effectively "braking" the flow to prevent burns. Our approach operates on a fundamentally inverse intuition: in unguided or standard regimes, models suffer from "undershooting" rather than saturation. Therefore, instead of suppressing energy at the end, SSC injects energy at the start to compensate for integration lag. This highlights that velocity errors are time-dependent and asymmetric: we need acceleration at launch, not just stability at landing.

## 3. Analysis of the Velocity Deficit

In this section, we first introduce several key concepts and then analyze the observed physics of the velocity deficit.

### 3.1. Flow Matching and the Velocity Deficit

**The MSE Objective and Optimal Solution.** While structured couplings (e.g., mini-batch OT(Pooladian et al., 2023; Tong et al., 2024), 2-Rectified Flow) aim to restrict or eliminate trajectory crossings to minimize conditional variance, standard Flow Matching models (e.g., SiT, REPA) are practically trained using Independent Data Coupling. Our work operates under this standard paradigm, formalized by the following assumption:

**Assumption 3.1.** The joint distribution factorizes as $q(x_0, x_1) = p_0(x_0)p_1(x_1)$, with non-degenerate marginals.

Standard Flow Matching trains a continuous normalizing flow by regressing a target vector field. The training objective is to minimize the Mean Squared Error (MSE) between the model prediction $v_\theta(x_t, t)$ and the target velocity $v_{\text{target}}$ over all possible pairs of data and noise:

$$\mathcal{L}_{\text{FM}} = \mathbb{E}_{t, x_0, x_1}\left[\|v_\theta(x_t, t) - v_{\text{target}}(x_t, t)\|^2\right], \quad (1)$$

where $v_{\text{target}} = \dot{\alpha}_t x_1 + \dot{\sigma}_t x_0$ defines the ground-truth trajectory from a specific noise $x_0$ to data $x_1$.

Since the training process optimizes this objective over the entire joint distribution $p(x_0, x_1)$, the global minimizer $v^*$ for any given input state $x_t$ is theoretically derived as the conditional expectation:

$$v^*(x_t, t) = \operatorname{argmin}_v \mathcal{L}_{\text{FM}} = \mathbb{E}\left[v_{\text{target}} \mid x_t\right]. \quad (2)$$

This implies that the model learns to predict the *average* velocity of all possible trajectories passing through $x_t$.

**The Ambiguity of Intermediate States.** Crucially, for any intermediate time $t \in (0, 1)$, the mapping from state $x_t$ to its boundaries $(x_0, x_1)$ is not bijective. A single point $x_t$ can be visited by infinite combinations of start points $x_0$ and end points $x_1$. Consequently, the target velocity at $x_t$ is not a fixed value but follows a distribution $p(v_{\text{target}}|x_t)$ with non-zero variance.

**The Velocity Deficit (Mathematical Proof).** Due to this inherent ambiguity, the learned velocity $v^*$ inevitably suffers from magnitude contraction. Crucially, because the target velocity follows a distribution $p(v_{target}|x_t)$ with non-zero variance under independent coupling, and the squared norm function $||\cdot||^2$ is strictly convex, applying Jensen's Inequality guarantees that the inequality holds strictly. We verify that the squared norm of the expectation (learned energy) is strictly less than the expectation of the squared norm (target energy):

$$\underbrace{||v^*(x_t)||^2 = ||\mathbb{E}[v_{target}|x_t]||^2}_{\text{Learned}} < \underbrace{\mathbb{E}[||v_{target}||^2|x_t]}_{\text{Ideal Target}}. \quad (3)$$

This strict inequality mathematically proves that the learned field $v^*$ systematically underestimates the kinetic energy required to transport probability mass.

### 3.2. Decomposing the Deficit: Two Regimes

We analyze this contraction behavior at the trajectory boundaries, where the coupling between $x_0$ and $x_1$ breaks down. In high-dimensional latent spaces, the components $x_1$ and $x_0$ are approximately orthogonal(Ledoux, 2001). As a result, the squared norm of the target velocity decomposes as

$$\begin{aligned}||v_{\text{target}}(x_t, t)||^2 &= ||\dot{\alpha}_t x_1 + \dot{\sigma}_t x_0||^2 \\ &\approx \dot{\alpha}_t^2||x_1||^2 + \dot{\sigma}_t^2||x_0||^2.\end{aligned} \quad (4)$$

Under Assumption 3.1, the omission of the conditional cross-term $\langle x_1, x_0 \rangle \mid x_t$ is justified by its vanishing behavior at the boundaries ($t \to 0, 1$) and its negligible magnitude in the interior, see Appendix subsection D.1.

**Early Stage ($t \to 0$): The "Start-Up" Problem.** At the exact boundary $t \to 0$, the boundary conditions define $\alpha_0 = 0$ and $\sigma_0 = 1$, so the model observes pure noise $x_0$. The target velocity is $v_{target} = \dot{\alpha}_0 x_1 + \dot{\sigma}_0 x_0$. Under Assumption 3.1 (Independent Data Coupling), $x_0$ reveals no information about $x_1$. Since the data distribution is normalized to have a zero mean ($\mathbb{E}[x_1] = 0$), the conditional expectation strictly converges: $\lim_{t \to 0} \mathbb{E}[x_1|x_t] = \mathbb{E}[x_1] = 0$. The optimal prediction becomes the asymptotic limit:

$$\lim_{t \to 0} v^*(x_t, t) = \lim_{t \to 0} (\dot{\alpha}_t \mathbb{E}[x_1|x_t] + \dot{\sigma}_t x_0) = \dot{\sigma}_0 x_0. \quad (5)$$

Comparing the norms via Eq. (4), we see a clear deficit:

$$\lim_{t \to 0} ||v^*||^2 = \dot{\sigma}_0^2||x_0||^2 < \dot{\alpha}_0^2||x_1||^2 + \dot{\sigma}_0^2||x_0||^2. \quad (6)$$

*Missing Signal Energy*

**Late Stage ($t \to 1$): The "Landing" Problem.** As $t \to 1$, we have $\alpha_1 = 1$ and $\sigma_1 = 0$, so the model observes data $x_1$. The target velocity is $v_{target} = \dot{\alpha}_1 x_1 + \dot{\sigma}_1 x_0$. By symmetry, since the initial noise $x_0$ is independently coupled, $\lim_{t \to 1} \mathbb{E}[x_0|x_t] = \mathbb{E}[x_0] = 0$. The optimal prediction becomes:

$$\lim_{t \to 1} v^*(x_t, t) = \lim_{t \to 1} (\dot{\alpha}_t x_1 + \dot{\sigma}_t \mathbb{E}[x_0|x_t]) = \dot{\alpha}_1 x_1. \quad (7)$$

Similarly, the norm is strictly smaller than the target:

$$\lim_{t \to 1} ||v^*||^2 = \dot{\alpha}_1^2||x_1||^2 < \dot{\alpha}_1^2||x_1||^2 + \dot{\sigma}_1^2||x_0||^2. \quad (8)$$

*Missing Noise Energy*

### 3.3. Empirical Analysis of Velocity Statistics.

To further validate our theoretical analysis of the velocity deficit, we visualize the statistics of the predicted velocity norm across different model scales in Figure 2. We observe a clear stratification of the mean velocity norm based on model capacity. Larger models (e.g., SiT-XL/2) consistently maintain a higher velocity magnitude throughout the trajectory compared to smaller counterparts (e.g., SiT-S/2). This indicates that in practice, beyond the inherent information loss discussed, the limited expressivity of the network acts as an additional bottleneck, hindering the full recovery of the conditional signal energy. Regardless of model size, all curves exhibit a sharp decline in magnitude as $t \to 1$. This confirms that the **velocity deficit is a structural property of the MSE objective rather than a capacity issue**. Besides, the velocity profile of the XL/2 model trained for 7M steps (yellow curve) is nearly identical to that of the 400k-step version (green curve). This suggests that the **velocity deficit is a converged behavior rather than a transient issue caused by insufficient training**. Simply extending the training duration yields diminishing returns in closing the energy gap.

### 3.4. The Asymmetry of Intervention

Although the MSE contraction (velocity deficit) occurs at both boundaries, its physical implications are fundamentally asymmetric due to the different roles of signal and noise:

1. **At $t = 1$, Contraction is Beneficial (Implicit Denoising):** Here, the target velocity component $\dot{\sigma}_1 x_0$ represents the specific realization of the initial noise. Since $x_0$ is irretrievable from the data $x_1$, the model learns to average out this term. Crucially, this contraction is desirable: it prevents the ODE from overfitting to the stochastic noise of the training paths. By

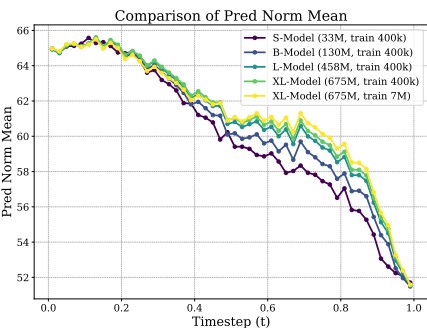

*Figure 2.* **Verification of Velocity Deficit.** We plot the mean velocity magnitude across timesteps. Regardless of model size or training duration (even 7M steps), all models exhibit a systematic monotonic decay, deviating from the constant target norm.

ignoring the $\dot{\sigma}_1 x_0$ component, the model effectively learns a smoother vector field that points directly to the data manifold, improving the FID(Heusel et al., 2017) score.

2. **At $t = 0$, Contraction is Harmful (Signal Starvation):** Here, the target velocity component $\dot{\alpha}_0 x_1$ represents the data signal that guides the noise towards the manifold. Due to the orthogonality of high-dimensional noise, the model fails to predict the signal component, outputting an initial velocity driven almost entirely by the noise term. Unlike the end time, this loss is catastrophic: the particle is "blind" to the data location at the very beginning, leading to a significant integration lag that cannot be recovered later.

**Conclusion:** To correct the Integration Lag without compromising the beneficial denoising property at the end, we must inject energy *non-uniformly*. This observation motivates our two methods with initial energy injection, restoring Signal at the start while respecting Denoising at the end.

## 4. Method

We propose two complementary strategies to address the velocity deficit: a training-based objective and a training-free inference correction.

### 4.1. Training: Magnitude-Aware Flow Matching (MAFM)

Standard FM loss entangles directional accuracy with magnitude regression. To address the early-stage deficit during training, we propose **Magnitude-Aware Flow Matching (MAFM)**, which augments the standard objective with an explicit magnitude supervision term:

$$\mathcal{L}_{\text{MAFM}} = \mathcal{L}_{\text{FM}} + \lambda(t) \cdot (\|v_\theta(x_t, t)\| - \|x_1 - x_0\|)^2 \quad (9)$$

We define the weighting schedule as

$$\lambda(t) = \lambda_0 \cdot (1 - t),$$

where $\lambda_0$ is a hyperparameter balancing the magnitude regularization strength (empirically set to $\lambda_0 = 0.2$). The weighting schedule is designed to enforce magnitude constraints strictly at $t \to 0$ (where signal starvation is critical) while relaxing them at $t \to 1$ to allow the beneficial velocity contraction (implicit denoising) to take effect.

### 4.2. Inference: Scale Schedule Corrector (SSC)

For pre-trained models where retraining is not feasible, we introduce the **Scale Schedule Corrector (SSC)**. SSC is a training-free, "plug-and-play" intervention that rectifies the velocity field $v_\theta$ using a time-dependent scaling factor $\gamma(t)$:

$$\hat{v}(x, t) = \gamma(t) \cdot v_\theta(x, t), \quad \text{where } \gamma(t) = s_{\text{start}} \cdot (1 - t) + s_{\text{end}} \cdot t \quad (10)$$

Based on our analysis of the asymmetric deficit:

- We set $s_{\text{start}} > 1.0$ (typically 1.1) to inject energy and compensate for integration lag.

- We set $s_{\text{end}} = 1.0$ to respect the natural convergence of the vector field, preserving high-frequency details.

**Implementation.** SSC requires only a single line of code modification in standard solvers (e.g., Euler, Heun). A PyTorch implementation is provided in Listing 1.

*Listing 1.* SSC requires minimal modification to the standard solvers.

```
def step_euler(model, x, t, dt):
    v = model(x, t)
    # Standard update:
    # return x + v * dt

    # SSC Update (Ours):
    # s_start=1.1, s_end=1.0
    gamma = torch.lerp(1.1, 1.0, t)
    return x + v * gamma * dt
```

**Empirical Rationale for the Linear Schedule**. We emphasize that the specific formulation of the Scale Schedule Corrector ($\gamma(t)$) is empirically driven rather than analytically optimal. Our theoretical derivation proves the necessity of Initial Energy Injection to resolve the velocity deficit, but the exact temporal decay shape is heuristic. While alternative non-linear profiles (e.g., Quad-In) can marginally yield lower FID scores (as detailed in Appendix subsection B.1), the linear schedule is selected as the default recommendation due to its simplicity. It serves as a minimal, robust "plug-and-play" baseline that provides the optimal trade-off between global structure repair and local texture preservation across diverse model families (including SiT, MMDit, and standard U-Net).

# 5. Experiments

## 5.1. Setup

**Datasets.** We evaluate our method on both class-conditional and text-to-image (T2I) generation. For class-conditional generation, we conduct experiments on ImageNet-1k (Deng et al., 2009) at $256 \times 256$ and $512 \times 512$ resolutions, following the preprocessing protocol of ADM (Dhariwal & Nichol, 2021). For T2I generation, we use MS-COCO (Lin et al., 2014) and adopt the data preprocessing and evaluation split setup of U-ViT (Bao et al., 2023; Yu et al., 2024), where performance is measured on the validation split.

**Implementation.** We train vanilla Scalable Interpolant Transformers (SiT) (Ma et al., 2024) at multiple scales (S/2, B/2, L/2, XL/2) under identical hyperparameter settings. The only differences are whether our proposed MAFM training strategy is applied and whether the SSC module is enabled at inference time. To assess the generalizability of our approach, we further integrate our method with REPresentation Alignment (REPA) (Yu et al., 2024) and train REPA-enhanced models (B/2, XL/2) on ImageNet-1k at $256 \times 256$ and $512 \times 512$ resolutions. REPA accelerates training and improves generative quality by aligning intermediate representations with pretrained vision encoders (e.g., DINOv2 (Oquab et al., 2023)) via an auxiliary distillation loss. For T2I generation, we train REPA-MMDiT (Esser et al., 2024) from scratch on MS-COCO following standard protocols. **Metrics.** For the formal definitions of evaluation metrics (FID, sFID, Inception Score, Precision, Recall), please refer to Appendix C.

## 5.2. Main Results & Effectiveness

### 5.2.1. DIRECT MEASUREMENT OF INTEGRATION LAG

To explicitly quantify the trajectory stalling, we track the Fréchet Latent Distance (FLD) in the VAE(Variational Autoencoder) latent space across sampling timesteps. Tracking FLD in the exact operational domain overcomes the concentration of measure challenges inherent in high-dimensional nearest-neighbor metrics.

As illustrated in Figure 3, measurements confirm that baseline particles systematically stall short of the data manifold. By applying SSC, particles are consistently pushed closer to the target distribution. The correction magnitude actively grows from the early steps (e.g., 3.6% closer at $t = 0.2$) to a significant peak in the mid-trajectory (e.g., over 28.2% closer at $t = 0.8$), directly proving the mitigation of integration lag.

Furthermore, endpoint verifications utilizing the Inception $NN - \ell_2$ distance demonstrate that generated images are fully denoised and maintain structural superiority at $t = 1$. For comprehensive FLD tracking data across all model

scales (SiT-S/2 through SiT-XL/2), please refer to **Appendix B.4**.

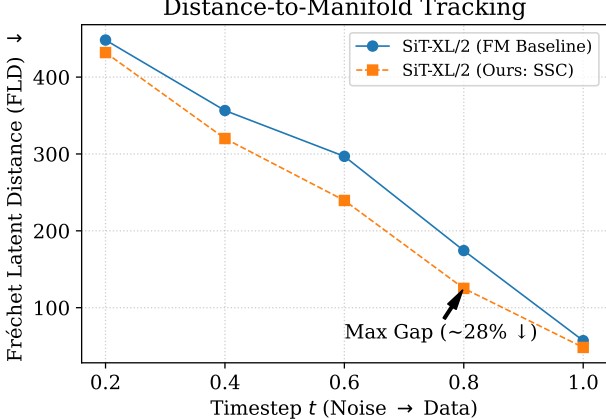

*Figure 3.* **Distance-to-Manifold Tracking.** Fréchet Latent Distance (FLD) measured across timesteps. Baseline models (dashed lines) exhibit significant integration lag. Our SSC (solid lines) consistently pulls the trajectory closer to the data manifold, achieving maximum structural correction at $t = 0.8$.

### 5.2.2. STANDARD BENCHMARKS

*Table 1.* **Quantitative Comparison on ImageNet-1k** $256 \times 256$. We evaluate both our training-based method (MAFM) and training-free method (SSC) across different model scales. Notably, the training-free SSC outperforms the training-based MAFM in FID while incurring zero training cost.

| Model | Method | FID ↓ | sFID ↓ | FID Imp. | sFID Imp. |
|-------|--------|-------|--------|----------|-----------|
| SiT-S/2 | Flow Matching | 64.83 | 16.00 | - | - |
| | + MAFM (Train) | 57.25 | 8.53 | +11.7% | +46.7% |
| | **+ SSC (Inference)** | **52.85** | **7.83** | **+18.5%** | **+51.1%** |
| SiT-B/2 | Flow Matching | 40.64 | 11.14 | - | - |
| | + MAFM (Train) | 35.83 | **6.12** | +11.8% | **+45.1%** |
| | **+ SSC (Inference)** | **32.72** | 6.46 | **+19.5%** | +42.0% |
| SiT-L/2 | Flow Matching | 23.18 | 8.79 | - | - |
| | + MAFM (Train) | 20.55 | **4.99** | +11.4% | **+43.2%** |
| | **+ SSC (Inference)** | **18.00** | 5.77 | **+22.3%** | +34.3% |
| SiT-XL/2 | Flow Matching | 18.70 | 8.09 | - | - |
| | + MAFM (Train) | 16.68 | **4.67** | +10.8% | **+42.3%** |
| | **+ SSC (Inference)** | **14.26** | 5.59 | **+23.7%** | +30.9% |

Table 1 provides a comprehensive comparison across model scales.

**1. Magnitude Supervision Works.** Comparing MAFM with the baseline (FM), the explicit magnitude loss yields consistent gains, improving **FID by 10.8% and sFID by over 42.3%** on SiT-XL/2. This confirms that the vanilla MSE objective indeed fails to capture sufficient energy for high-fidelity generation.

**2. Training-Free Correction outperforms Training-Based Methods.** Remarkably, our inference-time intervention (SSC) achieves superior FID scores compared to the

training-based MAFM (e.g., 14.26 vs. 16.68 on SiT-XL/2). This suggests that the Integration Lag is an intrinsic geometric property of the MSE-learned field that is easier to correct explicitly during sampling than to unlearn during training. By explicitly correcting the velocity field at inference time, SSC bypasses the inherent averaging bias of the training objective.

**3. Structure vs. Texture Trade-off.** We note an interesting dichotomy: while SSC excels in FID (global structure), MAFM often achieves slightly better sFID (local texture). We attribute this to the nature of the interventions: MAFM updates network weights to better model high-frequency details at the trajectory's end, whereas SSC focuses on correcting the global semantic layout at the trajectory's start.

### 5.2.3. ROBUSTNESS ON CONVERGED MODELS

*Table 2.* **Performance on Fully Converged Models with CFG.** SSC is evaluated with NFE = 50 on fully converged SiT-XL/2 and REPA SiT-XL/2 models. Across both unguided and guided settings, SSC consistently improves sample quality, demonstrating that velocity deficit persists beyond convergence and can be effectively corrected at inference time.

| Model | CFG | $s_{start} \rightarrow s_{end}$ | FID ↓ | sFID ↓ | IS ↑ |
|---|---|---|---|---|---|
| *SiT-XL/2 (7M steps)* | | | | | |
| Flow Matching | 1.0 | $1.0 \rightarrow 1.0$ | 13.68 | 11.38 | 111.18 |
| **+Ours (SSC)** | 1.0 | $1.1 \rightarrow 1.0$ | **7.58** | **5.12** | **138.07** |
| Flow Matching | 1.5 | $1.0 \rightarrow 1.0$ | 2.64 | 6.23 | 251.19 |
| **+Ours (SSC)** | 1.5 | $1.05 \rightarrow 1.0$ | **2.28** | **4.47** | 265.34 |
| **+Ours (SSC)** | 1.5 | $1.1 \rightarrow 1.0$ | 2.54 | 5.20 | **272.71** |
| *REPA SiT-XL/2 (4M steps)* | | | | | |
| Flow Matching | 1.0 | $1.0 \rightarrow 1.0$ | 10.60 | 10.30 | 132.9 |
| **+Ours (SSC)** | 1.0 | $1.1 \rightarrow 1.0$ | **5.99** | **5.42** | **154.96** |

**Validation on Fully Converged SOTA Models.** To rule out underfitting or insufficient optimization, we evaluate SSC on fully converged state-of-the-art models, including a SiT-XL/2 trained for 7M steps and a REPA SiT-XL/2 trained for 4M steps.

As shown in Table 2, SSC consistently improves sample quality across both architectures, despite full convergence. In the unguided setting, SSC substantially outperforms the baseline, indicating that **velocity deficit persists even after long training on SiT-XL/2 and is therefore a systematic bias of flow matching**.

**Complementarity with Classifier-Free Guidance.** SSC is a training-free method and is complementary to classifier-free guidance rather than a replacement. While classifier-free guidance modulates conditional strength, SSC corrects the sampling trajectory by adjusting the velocity field, operating on a distinct aspect of the generation process. As shown in Table 2, **SSC consistently improves performance under both unguided and guided settings.** The

gains persist even under strong guidance, indicating that SSC addresses deficiencies in the underlying flow dynamics that are not resolved by guidance alone.

Moreover, the observed gains are not highly sensitive to the choice of the initial scale. The same schedule ($s_{start} = 1.1 \rightarrow s_{end} = 1.0$) yields consistent improvements across models and guidance settings, suggesting that SSC provides a robust trajectory-level correction rather than relying on finely tuned hyperparameters.

Visual comparisons in **Appendix A.2** (Figure 7) further reveal that baseline samples from converged models still suffer from broken geometries, which SSC effectively restores.

### 5.2.4. EFFICIENCY VS. QUALITY

**Efficiency Analysis** To evaluate the sampling efficiency, we plot the FID scores against the NFE in Figure 4. The results highlight the efficiency gain from our method. As illustrated, our method with NFE=50 achieves an FID of 7.58, which **reduces FID by 44.6% with NFE=50** and also **outperforms the baseline with NFE=250** (FID 8.65). This implies that SSC allows for a **5× reduction in inference cost** while maintaining superior generation quality.

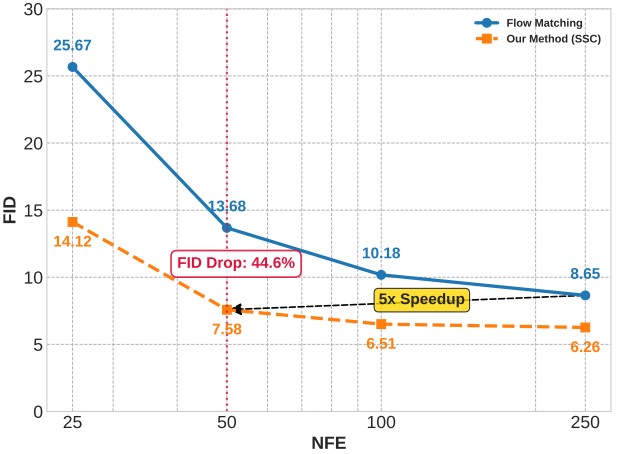

*Figure 4.* **Efficiency Analysis on ImageNet-1k** $256 \times 256$. SSC achieves the baseline's 250-step performance in just 50 steps (5× Speedup).

### 5.3. Generalization Capabilities

**Compatibility with REPA** Table 3 reports results of REPA-SiT backbones evaluated under a fixed 50-NFE budget at $256 \times 256$ and $512 \times 512$ resolutions. Under the REPA objective, our methods consistently improve generation quality over baselines. At 256×256 resolution, the training-based MAFM achieves **15.6%–21.8% FID reduction and 30.2%–42.4% sFID reduction**, while largely preserving Recall.

**Crucially, these gains extend to higher resolutions.** At

512×512, our training-free SSC reduces FID by **20.7%– 28.4%** (e.g., from 11.72 to 8.39 on SiT-XL/2), demonstrating that the velocity deficit is a scale-invariant phenomenon that can be effectively corrected even in high-resolution regimes.

*Table 3.* **Generalization to REPA and higher resolutions.** Our training-based (MAFM) and training-free (SSC) methods generalize to the REPA objective and consistently improve REPA-SiT on ImageNet-1k, including at $512 \times 512$ resolution.

| | (a) ImageNet-1k $256 \times 256$ | | | | |
|---|---|---|---|---|---|
| **Model** | **Method** | **FID** | **sFID** | **Precision** | **Recall** |
| SiT-B/2 | REPA | 28.26 | 11.83 | 0.57 | 0.64 |
| | **+Ours (MAFM)** | 23.84 | 6.81 | 0.59 | **0.65** |
| | **+Ours (SSC)** | **22.16** | **6.02** | **0.60** | 0.64 |
| SiT-XL/2 | REPA | 10.89 | 8.25 | 0.68 | **0.66** |
| | **+Ours (MAFM)** | 8.52 | 5.76 | 0.70 | 0.64 |
| | **+Ours (SSC)** | **7.69** | **5.35** | **0.71** | 0.64 |
| | (b) ImageNet-1k $512 \times 512$ | | | | |
| **Model** | **Method** | **FID** | **sFID** | **Precision** | **Recall** |
| SiT-B/2 | REPA | 31.73 | 13.76 | 0.67 | 0.61 |
| | **+Ours (SSC)** | **25.15** | **6.32** | **0.71** | **0.63** |
| SiT-XL/2 | REPA | 11.72 | 10.40 | 0.76 | **0.62** |
| | **+Ours (SSC)** | **8.39** | **5.18** | **0.79** | 0.61 |

**Text-to-Image Generation on MS-COCO dataset.** We evaluate our method on MS-COCO text-to-image generation to assess generalization beyond class-conditional settings. As shown in  Table 4, applying SSC to MMDiT trained with the REPA objective reduces FID **from 6.03 to 4.71**. These results indicate that the Velocity Deficit also manifests in text-conditioned generation, suggesting it is a general phenomenon in flow matching.

*Table 4.* **Text-to-Image Generation on MS-COCO.** FID comparison using the REPA-MMDiT backbone (NFE=50, CFG=1.0). SSC significantly reduces FID, showing generalization to text-conditioned generation.

| Method | FID |
|---|---|
| MMDiT + REPA | 6.03 |
| **+Ours (SSC)** | **4.71** |

**Improved Text Rendering on SANA.** Beyond standard metrics, we qualitatively demonstrate that SSC repairs structural defects in text rendering. As shown in Appendix A.1 (Figure 6), SSC corrects character misspellings (e.g., 'Homor' → 'Honor') in the state-of-the-art SANA model, confirming that integration lag compromises high-frequency semantic structures.

**Cross-Architecture Stability.** The velocity contraction bias is not exclusive to Transformer-based architectures. As detailed in Appendix B.1 (Table 8), applying the identical linear SSC to a 76M parameter CNN-based U-Net improves FID by 18.2% (from 57.10 to 46.73), confirming that inte-

*Table 5.* **Ablation on Scale Schedule.** Comparison of different injection schedules on the fully converged SiT-XL/2 model (ImageNet-1k $256 \times 256$) with NFE = 50. The proposed **Initial Energy Injection** ($1.1 \to 1.0$) yields the best FID by correcting the early-stage deficit. Constant scaling ($1.05 \to 1.05$) achieves the best sFID but slightly degrades FID, validating the structure-texture trade-off.

| $s_{start}$ | $s_{end}$ | **Method** | **FID ↓** | **sFID ↓** | **IS ↑** |
|---|---|---|---|---|---|
| 1.0 | 1.0 | Flow Matching | 13.68 | 11.38 | 111.18 |
| 1.0 | 1.1 | +Ours (SSC) | 8.36 | 5.69 | 130.64 |
| **1.1** | **1.0** | **+Ours (SSC)** | **7.58** | 5.12 | **138.07** |
| 1.05 | 1.05 | +Ours (SSC) | 7.75 | **4.93** | 134.81 |

gration lag is a universal MSE objective flaw.

### 5.4. Ablation study

**Injection Schedule.** We evaluate different schedules of $\gamma(t)$ on the **fully converged** SiT-XL/2 model to rigorously test the *Initial Injection* hypothesis (Table 5). The baseline schedule ($1.0 \to 1.0$) suffers from significant integration lag, yielding an FID of 13.68. While introducing injection only at the late stage ($1.0 \to 1.1$) improves performance (FID 8.36), it still lags behind the early injection strategy. This indicates that late-stage correction acts merely as a palliative measure—it cannot fully recover the cumulative trajectory deviation accrued during the signal-starved initial steps.

Applying a constant injection ($1.05 \to 1.05$) yields the best texture metrics (sFID 4.93). However, forcing energy injection until the terminal stage disrupts the natural velocity contraction, leading to a slightly degraded FID (7.75) compared to the optimal setting.

Our proposed early injection schedule ($1.1 \to 1.0$) achieves the best overall trade-off, obtaining the lowest FID of **7.58** while maintaining a competitive sFID of 5.12. By correcting the velocity deficit at the beginning (t→0) and annealing to identity at the end (t→1), this schedule effectively restores global structure without compromising the delicate denoising process required for fine details.

Qualitative results in Figure 5 corroborate these metrics. **Crucially, we observe that visual improvement is predominantly driven by increasing $s_{\mathbf{start}}$**, which repairs global structural defects (rows). In contrast, boosting $s_{\mathrm{end}}$ (columns) tends to over-smooth details. Consequently, our schedule ($1.1 \to 1.0$) strikes the optimal balance.

While our main paper focuses on the Linear schedule for its simplicity, we explore the impact of **diverse temporal energy distributions (e.g., Consine/Quad In/Quad Out) in Appendix subsection B.1**, providing deeper insights into the timing of energy injection.

**Flow Matching Path.** We investigate whether our method

$s_{\text{end}} = 1$     $s_{\text{end}} = 1.05$     $s_{\text{end}} = 1.1$

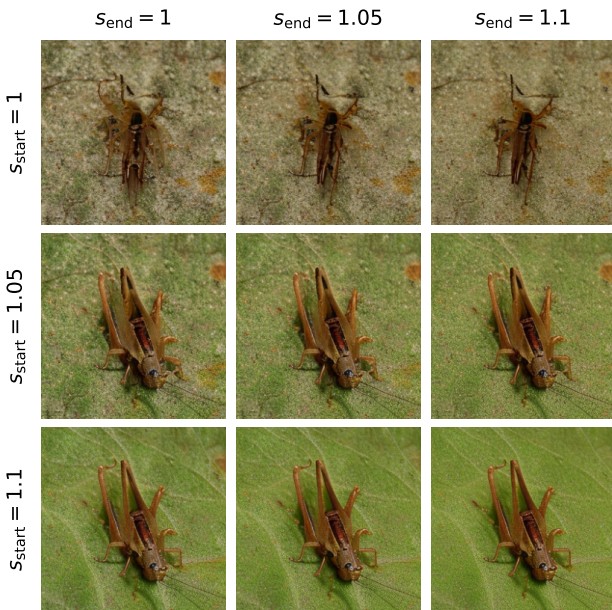

*Figure 5.* **Visual Ablation of Start vs. End Scaling ($s_{\text{start}}$ vs. $s_{\text{end}}$).** This grid visualizes the asymmetric impact of energy injection. **(1) Vertical Impact (Structural Repair):** Moving down the rows, increasing the initial scale $s_{\text{start}}$ from 1.0 to 1.1 significantly repairs the *Integration Lag*. Note how the baseline (Top Row) suffers from broken limbs and blurred boundaries, while the bottom row ($s_{\text{start}} = 1.1$) exhibits structurally coherent geometry. **(2) Horizontal Impact (Texture Fidelity):** Moving across columns, increasing the terminal scale $s_{\text{end}}$ tends to over-smooth high-frequency details (Right Column), while keeping $s_{\text{end}} = 1.0$ (Left Column) preserves crisp textures. **Conclusion:** Our method ($1.1 \rightarrow 1.0$, Bottom-Left) achieves the optimal trade-off, correcting structural defects without compromising textural sharpness.

is sensitive to the choice of flow matching path by evaluating SSC under different interpolants. As shown in Table 6, SSC consistently improves generation quality across all tested paths, indicating that the proposed method is largely path-agnostic. Specifically, applying SSC leads to substantial reductions in both FID and sFID for linear interpolation, SBDM-VP, and GVP paths, while maintaining or slightly improving Precision and Recall. These results suggest that the effectiveness of SSC does not rely on a particular flow construction, but rather on its ability to correct early-stage trajectory mismatch in a path-independent manner.

**Combining MAFM and SSC.** We detail these combination experiments in Appendix subsection B.2. While inference-based SSC physically forces the solver closer to the manifold optimizing macroscopic geometry (e.g., reaching an FID of 14.26 on SiT-XL/2), the training-based MAFM structurally updates spatial feature extractors, developing richer high-frequency gradients (e.g., reaching an sFID of 4.67). Stacking MAFM with a strong SSC can lead to kinetic energy oversaturation on larger models, degrading high-frequency delicate manifolds. Thus, MAFM is recommended for fidelity-critical texture synthesis, while SSC

*Table 6.* **Flow Matching Path Ablation.** Comparison of different flow matching interpolants with and without our proposed SSC. Our method consistently improves generation quality across all paths, indicating its geometric correction is path-agnostic.

| Interpolant | FID | sFID | Precision | Recall |
|---|---|---|---|---|
| Linear | 40.64 | 11.14 | 0.51 | **0.62** |
| **+Ours (SSC)** | **32.72** | **6.46** | **0.53** | **0.62** |
| SBDM-VP | 44.78 | 10.88 | 0.49 | 0.63 |
| **+Ours (SSC)** | **39.68** | **6.79** | **0.50** | **0.64** |
| GVP | 38.83 | 10.71 | 0.52 | 0.63 |
| **+Ours (SSC)** | **32.22** | **5.63** | **0.54** | **0.64** |

remains the optimal choice for compute-constrained deployment.

## 6. Limitations

While our methods effectively mitigate the integration lag, we identify the following limitations:

**Dependence on High-Dimensional Orthogonality.** Our theoretical derivations rely heavily on high-dimensional concentration of measure. Applying initial energy injection to ultra-low-dimensional toy datasets may cause the solver to overshoot the target manifold, potentially amplifying structural artifacts rather than repairing them.

**Hyperparameter Sensitivity in High-CFG Regimes.** SSC operates robustly across natural image and text-to-image tasks under standard or low CFG. However, in extremely high-CFG regimes (e.g., $CFG \geq 4.0$ in Appendix subsection B.4), the velocity field is already significantly amplified by the guidance scale. In such edge cases, our default $s_{start}$ may cause over-saturation, requiring empirical grid search to re-calibrate the initial injection scale.

## 7. Conclusion

In this work, we identified the Velocity Deficit not as a mere training artifact, but as a structural bias of the MSE objective in high-dimensional Flow Matching. We demonstrated that this deficit causes a systematic Integration Lag, preventing samples from reaching the data manifold. To resolve this, we proposed Initial Energy Injection. Our key insight—that velocity contraction is harmful at the start but beneficial at the end—led to two solutions: the training-based MAFM and the training-free SSC. Notably, SSC acts as a "free lunch" improvement: it delivers a $5\times$ speedup and better FID scores with zero retraining and one line of code. Our findings suggest that high-quality generation requires not just accurate trajectory directions, but also precise kinetic energy management. We believe this perspective opens new avenues for dynamics-aware sampling in continuous generative models.

## Impact Statement

This paper presents work whose goal is to advance the field of machine learning. There are many potential societal consequences of our work, none of which we feel must be specifically highlighted here.

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

# A. Qualitative Results

## A.1. Visual Comparison on the state-of-the-art text-to-image model

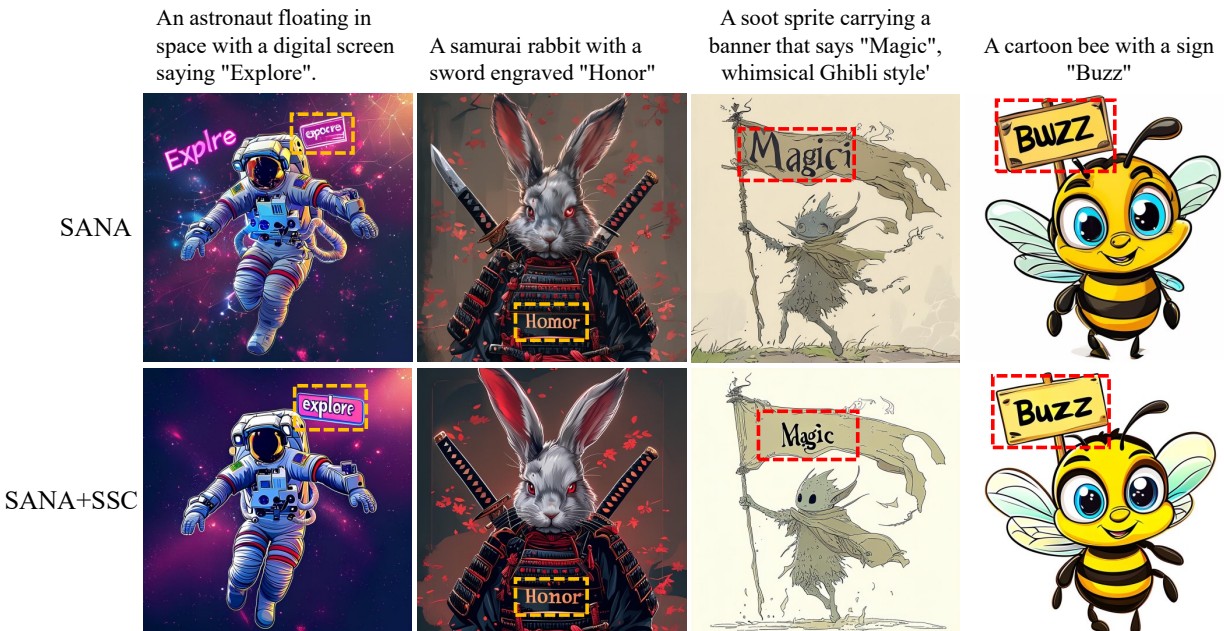

*Figure 6.* **Improved Text Rendering on SANA.** Text glyphs represent high-frequency manifolds. The baseline's integration lag causes "undershooting," resulting in blurred or incorrect characters. The baseline SANA (Top) fails to render the text accurately, e.g., producing "Homor". By applying our training-free SSC (Bottom), the model correctly renders "Honor", demonstrating that correcting the integration lag enables more precise convergence for high-frequency details like text glyphs.

**Case Study: Text Rendering on SANA.** To further verify the impact of SSC on fine-grained generation, we applied it to the state-of-the-art efficient model SANA (Xie et al., 2024). Text rendering is notoriously sensitive to trajectory precision, as slight integration errors can corrupt character structures.

As illustrated in Figure 6, given the prompt "*...engraved 'Honor'*", the baseline model suffers from structural ambiguity, misspelling the word as "Homor". In contrast, SANA equipped with SSC accurately generates the correct spelling "Honor". This suggests that by compensating for the velocity deficit, SSC ensures that the generation process lands precisely on the data manifold, effectively resolving high-frequency artifacts and improving character legibility.

## A.2. Qualitative Analysis and Visual Comparison on ImageNet-1k

To isolate the impact of the Velocity Deficit, we evaluate a fully converged SiT-XL/2 model (7M training steps) in Figure 7. Under the standard Flow Matching, we observe severe artifacts usually associated with early training stages:

1. **Structural Collapse:** Objects appear fragmented or fail to form closed contours.

2. **Incomplete Denoising:** Distorted local structure and high-frequency noise remains visible in texture regions.

These failures demonstrate that **Integration Lag creates a bottleneck that masks the model's true capability**. Even with a perfect vector field prediction, the cumulative underestimation of velocity prevents the sample from reaching the clean data manifold at $t = 1$.

By applying SSC, we effectively remove this bottleneck. As shown in the Figure 7, SSC restores structural integrity and sharpness, proving that the converged model *has* learned these details, but the standard solver simply failed to reach them.

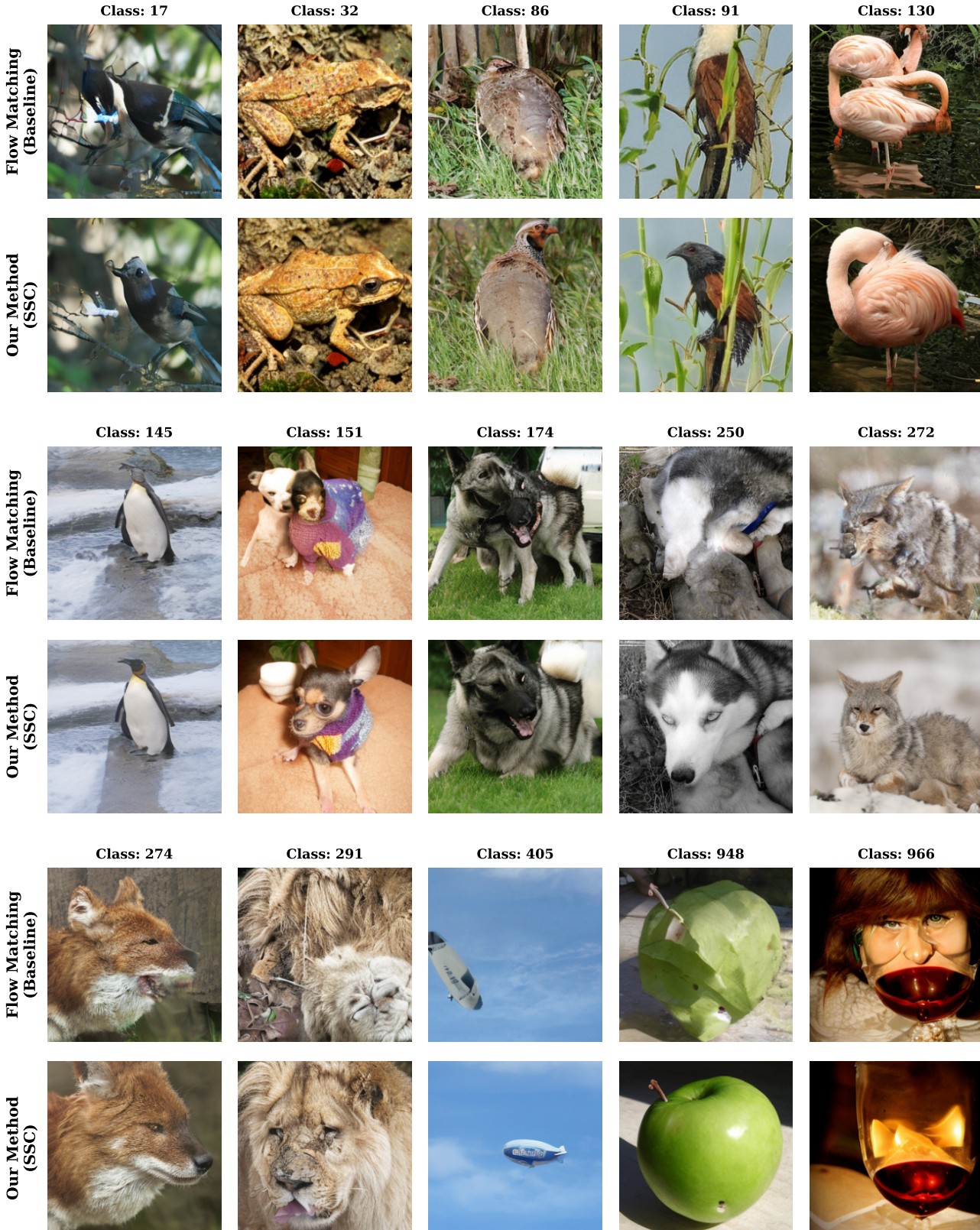

*Figure 7.* **Qualitative Comparison on Fully Converged Model.** Visual samples from a fully converged SiT-XL/2 model (trained for 7M steps on ImageNet-1k(256x256)) generated with NFE=50 and CFG=1.5. **Baseline:** The standard flow matching suffers from *Velocity Deficit*, causing the trajectory to stop short of the data manifold. This results in **incomplete objects**, **geometric distortions**, and **residual artifacts**—caused by **Integration Lag**. **Ours (+SSC):** By correcting the velocity magnitude, SSC enables the solver to fully traverse the trajectory, unlocking the true generation quality of the converged model without any retraining.

# B. Additional Analysis and Ablation Study

## B.1. Sensitivity to Schedule Shapes

To investigate whether the effectiveness of SSC stems from the specific linear trajectory or simply the total amount of injected energy, we conducted a **Schedule Shapes Analysis**.

We designed three alternative schedule shapes (Cosine, Quad In, Quad Out) alongside the standard Linear schedule. Crucially, to ensure a fair comparison, we adjusted the starting scale $s_{start}$ for each shape such that the **total injected energy** (defined as the area under the curve $\int_0^1 \gamma(t)dt$) remains approximately constant relative to the Linear baseline (Area $\approx 1.05$). The configurations are detailed in Table 7 and visualized in Figure 8.

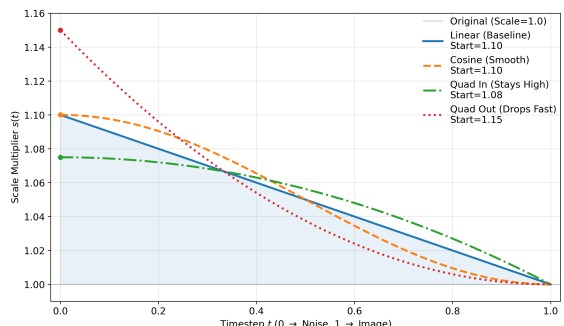

*Figure 8.* **Visualization of Scale Schedules.** Comparison of Linear, Cosine, Quad In, and Quad Out curves. All schedules are calibrated to provide strictly comparable total energy injection (Area Under Curve).

*Table 7.* **Performance of different schedule shapes.** Comparison of different decay profiles. **Quad In** performs best, suggesting sustained energy injection is beneficial. **Linear** is chosen as default for simplicity.

| Type | Profile | $s_{start}$ | $s_{end}$ | FID | sFID |
|------|---------|---------|-------|-----|------|
| **Linear** | Constant | 1.10 | 1.0 | 7.58 | 5.12 |
| Cosine | Smooth | 1.10 | 1.0 | 7.65 | 5.32 |
| **Quad In** | Sustain | 1.075 | 1.0 | **7.54** | **4.92** |
| Quad Out | Rapid | 1.15 | 1.0 | 7.76 | 5.68 |

The results in Table 7 reveal distinct performance characteristics governed by the temporal distribution of energy:

- **Quad In (Best Performance):** Surprisingly, the Quad In schedule ($s_{start} = 1.075$) slightly outperforms the Linear baseline, achieving the lowest FID (7.54) and sFID (4.92). Since the Quad In profile maintains scale values $> 1.0$ for a longer duration than Linear, sustaining the energy injection into the early-to-mid trajectory helps the solver better navigate these intermediate regions.

- **Quad Out (Worst Performance):** Despite a high initial boost ($s_{start} = 1.15$), the rapid decay causes performance to degrade (FID 7.76). This confirms that a momentary "kick" at the start is insufficient; the correction must be sustained to effectively counteract the integration lag.

- **Linear vs. Quad In:** While Quad In offers a marginal gain, we retain the **Linear** schedule as our default recommendation in the main paper. The performance gap is minimal ($\Delta$FID $= 0.04$), and the Linear schedule provides the simplest implementation that robustly solves the core problem.

**Cross-Architecture Stability on U-Net.** To formalize the stability of these empirical schedule shapes across entirely different model families, we conducted additional experiments applying Flow Matching to a standard U-Net architecture ($\sim$76M parameters, offering a capacity comparable to the range between SiT-S/2 and SiT-B/2). As Table 8 shows, the results align perfectly with the phenomena observed in the Transformer-based (SiT) models.

**Alternative schedules for MAFM weighting ($\lambda$).** Consistent with our findings on SSC schedule shapes, we tested alternative temporal decay profiles for the MAFM weighting schedule $\lambda(t)$ during training.

We calibrated the new profiles (Cosine, Quad In, Quad Out) such that the total integral of the penalty remained equal to our default linear strategy. As shown in Table 9 (evaluated on SiT-S/2), all profiles significantly improve over the baseline. While the "Quad In" profile offers a marginal gain in sFID, we retain the **Linear** schedule as our default recommendation because the performance differences are minimal, and the linear decay provides the simplest, most robust implementation.

*Table 8.* Performance comparison of Flow Matching on a standard U-Net architecture with various SSC schedules.

| Method | FID ↓ | sFID ↓ | IS ↑ | Precision ↑ | Recall ↑ |
|---|---|---|---|---|---|
| Baseline (U-Net Flow Matching) | 57.10 | 16.30 | 24.63 | 0.457 | 0.563 |
| **+ SSC (Linear, Default)** | 46.73 | **8.10** | **28.71** | 0.483 | **0.574** |
| + SSC (Cosine) | 46.99 | 8.58 | 28.62 | 0.479 | 0.573 |
| **+ SSC (Quad-In)** | **45.34** | 8.91 | 28.51 | **0.484** | 0.563 |
| + SSC (Quad-Out) | 49.43 | 9.54 | 27.92 | 0.475 | 0.570 |

*Table 9.* MAFM performance across different training penalty schedule shapes.

| Method & Shape Type | Profile Shape | FID (↓) | sFID (↓) |
|---|---|---|---|
| Flow Matching | Constant | 64.83 | 16.00 |
| **Linear MAFM** (default) | Linear Decay | 57.25 | 8.53 |
| Cosine MAFM | Smooth | 57.48 | 8.89 |
| **Quad In MAFM** | Sustain | **55.41** | **7.22** |
| Quad Out MAFM | Rapid | 60.07 | 10.76 |

## B.2. The MAFM+SSC combination and when to prefer which

We conduct the combination experiments across all SiT architectures. As shown in Table 10, there is a clear dichotomy between macroscopic global structure (measured by FID) and microscopic local texture (measured by sFID).

*Table 10.* Performance comparison among different methods across SiT architectures.

| Model | Method | FID (↓) | sFID (↓) |
|---|---|---|---|
| **SiT-S/2** | FM Baseline | 64.83 | 16.00 |
| | Our MAFM | 57.25 | 8.53 |
| | Our SSC | 52.85 | 7.83 |
| | Our MAFM+SSC ($s_{start}$=1.1) | **52.46** | 11.89 |
| | Our MAFM+SSC ($s_{start}$=1.05) | 53.75 | **7.75** |
| **SiT-B/2** | FM Baseline | 40.64 | 11.14 |
| | Our MAFM | 35.83 | **6.12** |
| | Our SSC | **32.72** | 6.46 |
| | Our MAFM+SSC ($s_{start}$=1.1) | 33.36 | 11.19 |
| | Our MAFM+SSC ($s_{start}$=1.05) | 33.54 | 6.43 |
| **SiT-XL/2** | FM Baseline | 18.70 | 8.09 |
| | Our MAFM | 16.68 | **4.67** |
| | Our SSC | **14.26** | 5.59 |
| | Our MAFM+SSC ($s_{start}$=1.1) | 16.30 | 10.62 |
| | Our MAFM+SSC ($s_{start}$=1.05) | 15.66 | 5.76 |

**The Trade-off Dynamics:**

- **SSC dominates FID (Global Structure):** Explicitly scaling the vector field during inference physically forces the solver closer to the manifold, optimizing macroscopic geometry (e.g., reaching 14.26 FID on SiT-XL/2).

- **MAFM dominates sFID (Local Texture):** Penalizing magnitude collapse during backpropagation structurally updates the spatial feature extractors, forcing the network to develop richer high-frequency gradients (e.g., reaching 4.67 sFID on SiT-XL/2).

- **The Combination:** Stacking MAFM and a strong SSC results in kinetic energy oversaturation, heavily degrading sFID

(10.62) by overshooting delicate high-frequency manifolds. However, a moderate combination successfully balances both metrics on smaller models (SiT-S/2).

**When is the training-based MAFM preferable?**

- **Compute-Constrained Deployment:** For already converged models, inference-based SSC is unequivocally optimal, offering massive structural repair with absolutely zero computational overhead.

- **High-Fidelity Textural Synthesis:** In specialized domains prioritizing local spatial consistency and crisp high-frequency textures (e.g., medical imaging or fine-grained generation), the training-based MAFM is essential to fundamentally update the network's spatial representations.

### B.3. Direct Measurement of Integration Lag.

Tracking the distance-to-manifold explicitly provides a direct measurement of the integration lag throughout the generation trajectory. To this end, we introduce the **Fréchet Latent Distance (FLD)**. We measure FLD specifically in the VAE latent space (the exact operational domain of the model) rather than relying on nearest-neighbor $\ell_2$ distance, which helps overcome the challenges of high-dimensional (4096-D) concentration of measure. For stable covariance estimation, we accumulate statistics over $8,192$ samples.

As shown in Table 11, baseline particles stall significantly short of the data manifold. In contrast, SSC-guided particles consistently remain closer across all timesteps. The correction magnitude actively grows from the early steps (e.g., $3.6\%$ closer at $t = 0.2$) to a peak in the mid-trajectory (e.g., over $28.2\%$ closer at $t = 0.8$), effectively quantifying the integration lag that SSC repairs. Furthermore, as a complementary endpoint verification, we measure the nearest-neighbor $\ell_2$ distance in the InceptionV3 feature space at the final step $t = 1$, where generated images are fully denoised and thus safely in-distribution for the InceptionV3 network. These unified results confirm the spatial superiority of the SSC trajectory.

*Table 11.* Distance-to-manifold tracking along the trajectory. Measurements include Fréchet Latent Distance (FLD) at intermediate timesteps and Inception NN-$\ell_2$ at the endpoint (SDE, 50 steps, 8,192 samples). Lower values indicate closer proximity to the data manifold.

| Model | Method | FLD @ $t = 0.2$ | FLD @ $t = 0.4$ | FLD @ $t = 0.6$ | FLD @ $t = 0.8$ | FLD @ $t = 1.0$ | Incep. NN-$\ell_2$ @ $t = 1$ |
|---|---|---|---|---|---|---|---|
| SiT-S/2 | FM | 448.8 | 358.8 | 300.3 | 178.2 | 62.9 | 15.6 |
| | SSC | 432.6 ($\downarrow$ 3.6%) | 322.8 ($\downarrow$ 10.0%) | 245.1 ($\downarrow$ 18.4%) | **133.0 ($\downarrow$ 25.4%)** | 53.5 ($\downarrow$ 15.0%) | **15.2 ($\downarrow$ 2.6%)** |
| SiT-B/2 | FM | 448.7 | 358.1 | 298.8 | 175.8 | 60.3 | 14.3 |
| | SSC | 432.5 ($\downarrow$ 3.6%) | 321.8 ($\downarrow$ 10.1%) | 242.2 ($\downarrow$ 19.0%) | **128.7 ($\downarrow$ 26.8%)** | 51.1 ($\downarrow$ 15.2%) | **13.8 ($\downarrow$ 3.0%)** |
| SiT-L/2 | FM | 448.39 | 356.99 | 297.47 | 174.50 | 57.88 | 12.99 |
| | SSC | 432.1 ($\downarrow$ 3.6%) | 320.9 ($\downarrow$ 10.1%) | 240.7 ($\downarrow$ 19.1%) | **126.5 ($\downarrow$ 27.5%)** | 48.8 ($\downarrow$ 15.7%) | **12.6 ($\downarrow$ 2.8%)** |
| SiT-XL/2 | FM | 448.2 | 356.5 | 296.9 | 174.3 | 57.1 | 12.6 |
| | SSC | 431.9 ($\downarrow$ 3.6%) | 320.1 ($\downarrow$ 10.2%) | 239.5 ($\downarrow$ 19.4%) | **125.1 ($\downarrow$ 28.2%)** | 48.3 ($\downarrow$ 15.5%) | **12.1 ($\downarrow$ 3.7%)** |

### B.4. Consistency in Text-to-Image Generation.

To demonstrate the generalization capability of the proposed method across different generative conditions, we evaluate the MMDiT+REPA backbone across multiple random seeds and Classifier-Free Guidance (CFG) scales. As shown in Table 12, SSC yields highly consistent performance gains with notably low variance across different seeds, particularly excelling in unguided and low-guidance regimes.

## C. Experimental Details

**Default Experimental Setting.** Unless otherwise specified, all experiments are conducted using models trained for 400K iterations, or equivalently, by performing training-free inference on the corresponding 400K-step checkpoints. To focus on the effect of geometric trajectory correction, we evaluate all methods without classifier-free guidance (i.e., CFG = 1.0) by default.

*Table 12.* Robustness of Text-to-Image generation (measured in FID) across various CFG scales and random seeds.

| Method | CFG Scale | Seed 0 | Seed 42 | Seed 123 | Mean | Std |
|---|---|---|---|---|---|---|
| MMDiT + REPA | 1.0 | 9.84 | 9.78 | 9.90 | 9.84 | 0.06 |
| SSC | 1.0 | 6.71 | 6.74 | 6.64 | **6.70** ($\downarrow$ **31.9%**) | 0.05 |
| MMDiT + REPA | 2.0 | 6.03 | 6.02 | 6.16 | 6.07 | 0.08 |
| SSC | 2.0 | 4.71 | 4.77 | 4.85 | **4.78** ($\downarrow$ **21.3%**) | 0.07 |
| MMDiT + REPA | 4.0 | 4.29 | 4.23 | 4.32 | 4.28 | 0.05 |
| SSC | 4.0 | 4.53 | 4.50 | 4.56 | 4.53 ($\uparrow$ 5.8%) | 0.03 |

**Metrics.** For class-conditional generation, we report five standard metrics computed on 50,000 generated samples: Fréchet Inception Distance (FID) (Heusel et al., 2017), Inception Score (IS) (Salimans et al., 2016), spatial FID (sFID) (Nash et al., 2021), Precision (Prec.), and Recall (Rec.) (Kynkäänniemi et al., 2019). For T2I generation, we report FID on the full validation set. Unless otherwise specified, we use the SDE Euler–Maruyama sampler with $w_t = \sigma_t$ and set the number of function evaluations (NFE) to 50.

**Robustness of hyperparameters** . $s_{start} = 1.1$ is simply searched on SiT-S/2 and then fixed. We found that $s_{start} = 1.1$ is robust across different architectures (SiT-S/2 to SiT-XL/2) and datasets (ImageNet to COCO), indicating it corrects a universal bias related to dimensionality rather than a specific dataset characteristic.

# D. Quantitative Proof

## D.1. Marginal vs. Conditional Independence (Eq. 4)

While the cross-term $\langle x_1, x_0 \rangle$ conditioned on $x_t$ might appear non-trivial, its omission is justified by its asymptotic behavior. In the following, we demonstrate that this term vanishes at the boundaries and remains negligible throughout the interior of the trajectory.

**Boundary Exactness** ($t \to 0, 1$)

Under independent coupling, $x_0$ reveals no information about $x_1$. With zero-mean preprocessed data ($\mathbb{E}[x_1] = 0$):

$$\lim_{t \to 0} \mathbb{E}[\langle x_1, x_0 \rangle \mid x_t] = \langle \mathbb{E}[x_1 \mid x_0], x_0 \rangle = \langle \mathbb{E}[x_1], x_0 \rangle = 0$$

By symmetry, this vanishes as $t \to 1$. Thus, the deficit at critical launch/landing phases (motivating our SSC) is unaffected.

**Asymptotic Negligibility in the Interior** ($t \in (0, 1)$)

To understand the interior dynamics, assuming a standard linear interpolant path under an isotropic Gaussian proxy ($x_1 \sim \mathcal{N}(0, \sigma_1^2 I_D)$, $x_0 \sim \mathcal{N}(0, I_D)$), the conditional expectation expands to:

$$\mathbb{E}\left[\langle x_0, x_1 \rangle \mid x_t\right] = (1-t)t \frac{\sigma_1^2}{\sigma_t^2} \left(\frac{\|x_t\|^2}{\sigma_t^2} - D\right)$$

By Measure Concentration, $\frac{\|x_t\|^2}{\sigma_t^2} \sim \chi_D^2$ tightly bounds the numerator to $\mathcal{O}(\sqrt{D})$. Since the principal kinetic energy (the denominator) scales linearly with dimension as $\Theta(D)$, the relative error from dropping the cross-term strictly converges to $\mathcal{O}(1/\sqrt{D})$. For $D = 4096$, this error is $\approx 1.56\%$.

**Empirical Verification on Real ImageNet Latents**

To verify this holds for real, non-Gaussian data without conditional estimation artifacts, we directly analyze the per-sample relative cross-term magnitude $\rho = 2|\langle x_0, x_1 \rangle|/(\|x_0\|^2 + \|x_1\|^2)$. Since $2|\langle x_0, x_1 \rangle| = \rho \cdot (\|x_0\|^2 + \|x_1\|^2)$ pointwise, taking $\mathbb{E}[\cdot \mid x_t]$ yields:

$$\frac{|2\mathbb{E}[\langle x_0, x_1 \rangle \mid x_t]|}{\mathbb{E}[\|x_0\|^2 + \|x_1\|^2 \mid x_t]} \leq \frac{\mathbb{E}[2|\langle x_0, x_1 \rangle| \mid x_t]}{\mathbb{E}[\|x_0\|^2 + \|x_1\|^2 \mid x_t]} \leq \sup \rho$$

Across 50K ImageNet VAE latent pairs ($D = 4096$), $\rho$ exhibits extreme measure concentration consistent with the $\mathcal{O}(1/\sqrt{D})$ high-dimensional limit:

- Mean $\rho = 1.24\%$

- 99th percentile $= 3.98\%$

- Absolute observed maximum $= 7.27\%$

Since $\rho < 4\%$ for over 99% of the sample space and never exceeds 7.3%, the conditional relative error from omitting the cross-term is bounded at this scale—confirming the approximation is tight regardless of distributional assumptions on $x_1$.

