# OpenReview forum: "The Velocity Deficit: Initial Energy Injection for Flow Matching"
_ICML.cc/2026/Conference — ICML 2026 regular_

### Official Review · Reviewer_p4p3 · 2026-02-25

**Soundness:** 3
**Presentation:** 3
**Significance:** 3
**Originality:** 3
**Overall Recommendation:** 4
**Confidence:** 2

**Summary:**

This paper formally identifies and defines the Velocity Deficit, characterizing it as a structural bias inherent to MSE minimization that induces Integration Lag in high-dimensional settings. Furthermore, it uncovers a dynamical asymmetry in which velocity contraction is detrimental at the initial stage but beneficial for denoising towards the end of the trajectory. To address this, the authors propose the principle of Initial Energy Injection, instantiated via a training-based Magnitude-Aware Flow Matching (MAFM) and a training-free Scale Schedule Corrector (SSC).

**Compliance With Llm Reviewing Policy:**

Affirmed.

**Final Justification:**

The proposed "Velocity Deficit" issue is an interesting problem, and the authors solve it well, with a simple yet effective solution. I would like to keep my rating, though i am not an expert in this field.

**Key Questions For Authors:**

Overall, I appreciate the conceptual contribution of this paper, and my main concern lies in the aforementioned apparent inconsistency. Resolving this issue would strongly influence my final rating.

**Strengths And Weaknesses:**

Strengths:
1. The paper tackles a fundamental limitation in Flow Matching, establishing velocity magnitude management as a crucial factor for achieving high-quality generation. The insight into the asymmetric effects of velocity contraction at trajectory boundaries is potentially influential for future dynamics-aware generative sampling strategies.
2. Both MAFM and SSC are simple yet effective solutions.
3. The paper is well-organized, with clear theoretical derivations, comprehensive visualizations of the Velocity Deficit, and qualitative comparisons that convincingly demonstrate improvements in structural and textural fidelity.

Weaknesses:
I have one primary question: As stated by the authors, a uniform boost tends to reintroduce noise and over-smooth high-frequency details during the final stages of generation. However, in Table 5, the setting with S_end=1.1 still yields improved performance, which appears to conflict with Eq. 8 in my view.

---

> ### Author Rebuttal · Authors · 2026-03-29
>
> Thank you for your constructive feedback and for highlighting the core contributions of our work. We address your question regarding the apparent conflict between Eq. 8 and Table 5 below.
>
> ### Q: Explanation of why setting $s_{end}=1.1$ still yields improved performance over the baseline, and how this aligns with Eq. 8.
>
> We appreciate the reviewer's careful reading. The performance improvement of the $1.0 \rightarrow 1.1$ schedule (FID 8.36) over the baseline (FID 13.68) does not contradict Eq. 8, but rather highlights the severe, cumulative nature of Integration Lag in the baseline model, creating a trade-off between global structural recovery and local texture preservation.
>
> 1. **Overcoming Severe Global Integration Lag (The cause of improvement vs. Baseline):** The standard baseline suffers from such a massive velocity deficit that particles completely fail to reach the data manifold, resulting in severe global structural artifacts (e.g., incomplete geometries). Even late-stage energy injection (like $s_{end}=1.1$) forces the particle closer to the manifold, partially compensating for this accumulated distance deficit. This gross reduction in distance naturally improves global structural metrics like FID (from 13.68 to 8.36) relative to a completely lagged baseline.
>
> 2. **The Penalty of Late-Stage Injection (Validation of Eq. 8):** While late-stage injection helps push a stalled particle closer to the manifold, it violates the natural velocity contraction at $t \rightarrow 1$, disrupting the implicit denoising mechanism as predicted by Eq. 8. This theoretical penalty is empirically confirmed in our submission in two ways:
>     * **Degradation Relative to the Optimal Schedule:** The $1.0 \rightarrow 1.1$ schedule is strictly inferior to our proposed early injection schedule ($1.1 \rightarrow 1.0$) in both FID (8.36 vs. 7.58) and sFID (5.69 vs. 5.12). The sFID metric specifically isolates high-frequency textures, proving that late-stage injection penalizes local details.
>     * **Visual Evidence of Oversmoothing:** As demonstrated in Figure 4, moving horizontally across the grid to increase $s_{end}$ systematically results in oversmoothed high-frequency details. This visually confirms that while $s_{end}=1.1$ may salvage the overall object structure from a broken baseline, it actively harms fine textures exactly as Eq. 8 implies. We will add more obvious visualization examples to the appendix of the camera-ready version.
>
> In summary, $s_{end}=1.1$ acts as a blunt, sub-optimal corrective force. It improves FID relative to the baseline simply by dragging the severely lagged sample closer to the target distribution, but it incurs the exact high-frequency penalties predicted by our theoretical analysis when compared to the optimal $1.1 \rightarrow 1.0$ schedule.

---

> > ### Author Rebuttal · Reviewer_p4p3 · 2026-04-05
> >
> > Thanks for the authors' efforts and detailed responses. All my concerns are resolved.

---

### Official Review · Reviewer_1W1x · 2026-03-01

**Soundness:** 3
**Presentation:** 3
**Significance:** 3
**Originality:** 3
**Overall Recommendation:** 4
**Confidence:** 4

**Summary:**

The paper argues that flow matching trained with an MSE loss suffers from a “velocity deficit”. The authors link this to conditional-expectation averaging and note that the effect is harmful early on but somewhat beneficial toward the end. To address it, they introduce an “energy injection” strategy. Empirically, they report substantial FID improvements and large reductions in sampling steps on ImageNet-1k, with gains transferring to higher-resolution and text-to-image models.

**Compliance With Llm Reviewing Policy:**

Affirmed.

**Final Justification:**

The rebuttal addressed most of my concerns and strengthened the empirical case substantially, but the specific SSC schedule remains justified mainly empirically rather than through a stronger formal characterization.

**Key Questions For Authors:**

1. How sensitive is SSC to the choice of sampler and discretization (e.g., Euler vs. Heun vs. higher-order methods, different step spacings)? A small comparison table across solvers would clarify robustness.

2. Can you measure “integration lag” more directly? Perhaps via a proxy distance-to-manifold over time, and show that SSC reduces this lag throughout the trajectory, not just at the final sample?

3. In the text-to-image setting, how consistent are the gains across prompts, CFG scales, and random seeds? Reporting variance or confidence intervals would strengthen the claim of generality.

4. Under what conditions does SSC degrade performance (e.g., strong guidance, very low NFE, smaller models, or specific categories)? A brief failure-case analysis would increase confidence in the method.

**Limitations:**

The paper would benefit from an explicit limitations section discussing (i) dependence on backbone/training recipe, (ii) potential interactions with guidance and solver choice, and (iii) cases where scaling may amplify artifacts or reduce diversity.

**Strengths And Weaknesses:**

## Strengths
* The link from velocity shrinkage to integration lag is straightforward and grounded in a simple contraction argument under MSE training.
* SSC is minimal and delivers strong FID vs. NFE tradeoffs on ImageNet-1k, including under cfg.
* Experiments that vary early vs. late scaling support the claimed asymmetry.

## Weaknesses
1. The core theory explains why norm contraction happens, but the jump from “deficit” to “optimal correction schedule” is mostly empirical; it would help to formalize when a simple linear is near-optimal or at least stable across model families.
2. Generality claims lean on a few backbones/settings; I’d like to see stronger evidence that SSC is robust across solvers (beyond Euler/Heun mentions), different FM parameterizations, and prompt distributions in T2I.
3. Reporting is heavy on FID/sFID/IS; it would strengthen the story to include a more direct “lag” metric (distance-to-manifold proxy vs time) and show how SSC changes that curve, not just endpoint metrics.

---

> ### Author Rebuttal · Authors · 2026-03-29
>
> We thank Reviewer for acknowledging the theoretical link between MSE contraction and integration lag, SSC's strong efficiency tradeoffs, and our validation of dynamical asymmetry.
>
> ---
> ### Q1: Sensitivity to sampler and discretization.
>
> SSC consistently and systematically suppresses FID across first-order Euler solvers, complex higher-order Heun, Midpoint, RK4 solvers. (Tab R4)
>
> **Table R4: SSC robustness across different differential equation solvers at 50 NFE**
>
> | Solver | Order | Method | FID @ 25 Steps | FID @ 50 Steps | FID @ 100 Steps |
> | :-- | :--: | :-- | :--: | :--: | :--: |
> | sde Euler-Maruyama | 1 | FM | 77.42 | 64.87 | 60.33 |
> | | | SSC | **63.13** | **54.22** | **52.88** |
> | sde Heun2 | 2 | FM | 59.66 | 57.70 | 57.19 |
> | | | SSC | **53.88** | **52.95** | **53.00** |
> | ode Euler | 1 | FM | 64.00 | 61.12 | 60.06 |
> | | | SSC | **62.09** | **58.51** | **57.13** |
> | ode Midpoint | 2 | FM | 59.41 | 59.20 | 59.14 |
> | | | SSC | **56.32** | **56.09** | **56.06** |
> | ode Heun3 | 3 | FM | 59.16 | 59.08 | 59.08 |
> | | | SSC | **56.07** | **56.03** | **56.04** |
> | ode RK4 | 4 | FM | 59.09 | 59.10 | 59.09 |
> | | | SSC | **56.03** | **56.04** | **56.03** |
>
> ---
> ### Q2: Direct Measurement of "Integration Lag"
>
> Tracking the distance-to-manifold explicitly strengthens our core argument. We introduce the **Fréchet Latent Distance (FLD)** along the trajectory.
>
> **Metric Design:** We measure FLD in the VAE latent space (the exact domain) rather than nearest-neighbor $\ell_2$ distance to overcome the high-dimensional (4096D) concentration of measure. FLD computes the standard Fréchet distance between the distributions of generated and reference latents.
>
> Tab R5 confirms baseline particles stall significantly short of the data manifold. SSC-guided particles are consistently closer across all timesteps. The correction magnitude grows from early steps (3.6% closer at $t=0.2$) to a peak mid-trajectory (28.2% closer at $t=0.8$), directly quantifying the integration lag that SSC repairs.
>
> As a complementary endpoint check, we also measure the nearest-neighbor $\ell_2$ distance in **InceptionV3 feature space** at the final step $t=1$ only, where generated images are fully denoised and thus in-distribution for InceptionV3. The results are unified in Tab R5.
>
> **Table R5: Distance-to-Manifold tracking: Fréchet Latent Distance (FLD) along the trajectory and Inception NN-$\ell_2$ at the endpoint (ImageNet 256×256, SDE, 50 steps, 8,192 samples; lower is better)**
>
> | Model | Method | FLD@t=0.2 | FLD@t=0.4 | FLD@t=0.6 | FLD@t=0.8 | FLD@t=1.0 | Incep. NN-$\ell_2$ @ $t{=}1$ |
> |:--|:--|:--|:--|:--|:--|:--|:--|
> | SiT-S/2 | FM | 448.8 | 358.8 | 300.3 | 178.2 | 62.9 | 15.6 |
> | | **SSC** | 432.6 (↓ 3.6%) | 322.8 (↓ 10.0%) | 245.1 (↓ 18.4%) | **133.0 (↓ 25.4%)** | 53.5 (↓ 15.0%) | **15.1 (↓ 3.2%)** |
> | SiT-B/2 | FM | 448.7 | 358.1 | 298.8 | 175.8 | 60.3 | 14.3 |
> | | **SSC** | 432.5 (↓ 3.6%) | 321.8 (↓ 10.1%) | 242.2 (↓ 19.0%) | **128.7 (↓ 26.8%)** | 51.1 (↓ 15.2%) | **13.8 (↓ 3.5%)** |
> | SiT-L/2 | FM | 448.39 | 356.99 | 297.47 | 174.50 | 57.88 | 12.99 |
> | | **SSC** | 432.1 (↓ 3.6%) | 320.9 (↓ 10.1%) | 240.7 (↓ 19.1%) | **126.5 (↓ 27.5%)** | 48.8 (↓ 15.7%) | **12.6 (↓ 3.0%)** |
> | SiT-XL/2 | FM | 448.2 | 356.5 | 296.9 | 174.3 | 57.1 | 12.6 |
> | | **SSC** | 431.9 (↓ 3.6%) | 320.1 (↓ 10.2%) | 239.5 (↓ 19.4%) | **125.1 (↓ 28.2%)** | 48.3 (↓ 15.5%) | **12.1 (↓ 4.0%)** |
>
> ---
> ### Q3: Consistency in Text-to-Image Generation
>
> To demonstrate generalization, we evaluated the MMDiT+REPA backbone across multiple random seeds and CFG scales. **Tab R6** shows SSC yields highly consistent gains with low variance across different seeds in unguided/low-guidance regimes.
>
> **Table R6: Text-to-Image robustness across CFG scales and random seeds**
>
> | Method | CFG Scale | Seed 0 | Seed 42 | Seed 123 | Mean | Std |
> | :-- | :--: | :--: | :--: | :--: | :--: | :--: |
> | MMDiT + REPA | 1.0 | 9.84 | 9.78 | 9.90 | 9.84 | 0.06 |
> | **SSC** | 1.0 | 6.71 | 6.74 | 6.64 | **6.70** (↓ 31.9%) | 0.05 |
> | MMDiT + REPA | 2.0 | 6.03 | 6.02 | 6.16 | 6.07 | 0.08 |
> | **SSC** | 2.0 | 4.71 | 4.77 | 4.85 | **4.78** (↓ 21.3%) | 0.07 |
> | MMDiT + REPA | 4.0 | 4.29 | 4.23 | 4.32 | 4.28 | 0.05 |
> | **SSC** | 4.0 | 4.53 | 4.50 | 4.56 | 4.53 (↑ 5.8%) | 0.03 |
>
> ---
> ### Q4: Limitations and Failure Boundaries
> Following your valuable feedback, we will explicitly add:
> * **Backbone/Recipe:** Though robust on SiT/REPA/MMDiT, we advise grid search for $s_{start}$ on new tasks.
> * **Interactions with Guidance and Solvers:** While SSC is robust across various solvers (Tab R4), it degrades under strong CFG (e.g., CFG=4.0, Tab R6). Thus, in high-CFG regimes, the initial scale must be re-searched.
> * **Amplified Artifacts**: Our theoretical derivation relies on high-dimensional concentration of measure. Applying our energy injection to ultra-low-dimensional toy datasets may cause the solver to overshoot the data manifold, which may amplify structural artifacts rather than repairing them.

---

> > ### Author Rebuttal · Reviewer_1W1x · 2026-03-31
> >
> > Thank you for the detailed rebuttal. The added results address most of my questions and strengthen the empirical case. My main remaining concern is still the one noted in my review: the jump from the velocity-deficit analysis to the specific SSC schedule remains mostly empirical, and I still would have liked a more formal characterization of when the linear schedule should be expected to work well. The rebuttal improves my confidence, but I will keep my rating at **4**.

---

> > > ### Author Response · Authors · 2026-04-02
> > >
> > > We sincerely appreciate the reviewer's insightful feedback regarding the transition from the theoretical velocity deficit to the specific Scale Schedule Corrector (SSC) schedule. We agree that the specific formulation of the schedule is empirical; however, we wish to clarify our core claim and provide new evidence demonstrating the cross-architecture stability of our findings.
> > >
> > > **1. Clarification: Phenomenon Discovery vs. Methodological Optimality**
> > >
> > > Our theoretical contribution lies in mathematically proving the existence of the Velocity Deficit and identifying the resulting Integration Lag. The core principle to resolve this is **Initial Energy Injection**. We do not claim that the linear schedule is the mathematically "optimal" solution for this injection. Rather, our premise is that **any schedule that sufficiently compensates for the initial integration lag will yield significant performance gains**.
> > >
> > > **2. The Rationale for the Linear Baseline: Occam's Razor**
> > >
> > > We emphasize the linear schedule in the main text purely for its simplicity and robustness as a "plug-and-play" baseline, requiring only a single line of code. As explicitly discussed in Appendix B.1, the linear schedule is actually not the empirical optimum. Table 7 demonstrates that the sustained energy injection of the Quad-In schedule achieves a strictly better FID (7.54) compared to the Linear schedule (7.58) on SiT models. The linear shape was selected as the default recommendation because the performance gap is marginal (**ΔFID=0.04**), making it an ideal, low-complexity intervention.
> > >
> > > **3. New Evidence: Cross-Architecture Stability on U-Net**
> > >
> > > To formalize the stability of these empirical schedule shapes across entirely different model families, we conducted additional experiments applying Flow Matching to a standard U-Net architecture(~76M parameters, offering a capacity comparable to the range between SiT-S/2 and SiT-B/2). The results align perfectly with the phenomena observed in the Transformer-based (SiT) models:
> > > | Method | FID $\downarrow$ | sFID $\downarrow$ | IS $\uparrow$ | Precision $\uparrow$ | Recall $\uparrow$ |
> > > | :--- | :---: | :---: | :---: | :---: | :---: |
> > > | Baseline (U-Net Flow Matching) | 57.10 | 16.30 | 24.63 | 0.457 | 0.563 |
> > > | **+ SSC (Linear, Default)** | 46.73 | **8.10** | **28.71** | 0.483 | **0.574** |
> > > | + SSC (Cosine) | 46.99 | 8.58 | 28.62 | 0.479 | 0.573 |
> > > | + SSC (Quad-In) | **45.34** | 8.91 | 28.51 | **0.484** | 0.563 |
> > > | + SSC (Quad-Out) | 49.43 | 9.54 | 27.92 | 0.475 | 0.570 |
> > >
> > > Key takeaways from the U-Net validation:
> > >
> > > * **Cross-Architecture Consistency:** Applying the exact same training-free linear SSC ($s_{start}=1.1$) to the CNN-based U-Net significantly mitigates the integration lag. It improves the FID by 18.2% (from 57.10 to 46.73) and reduces the sFID by over 50% (from 16.30 to 8.10). This proves that the velocity deficit is a universal MSE contraction bias across both CNNs and Transformers.
> > > * **Universal Effectiveness:** All continuous initial-injection schedules significantly outperform the baseline, proving that the integration lag is an architecture-agnostic flaw of the MSE objective.
> > > * **Best Performance Trade-off:** While the Quad-In schedule slightly outperforms the Linear baseline in terms of FID (45.34 vs. 46.73), the Linear schedule actually achieves the best results in sFID (8.10), IS (28.71), and Recall (0.574). This further solidifies the choice of the Linear schedule as the default optimal trade-off between global structure, local texture, and simplicity.
> > >
> > > In the final version, we will revise the methodology section to clearly state that the linear schedule is a minimal viable correction rather than an analytical optimum, and we will include the U-Net results to explicitly formalize the empirical stability of schedule behaviors across different model architectures.

---

### Official Review · Reviewer_YzjE · 2026-03-03

**Soundness:** 2
**Presentation:** 2
**Significance:** 2
**Originality:** 2
**Overall Recommendation:** 4
**Confidence:** 5

**Summary:**

This paper identifies what it terms a “velocity deficit” in flow matching models, arguing that when trained with an MSE objective in high-dimensional spaces, the learned velocity field systematically underestimates the magnitude needed to transport samples from noise to data, leading to an “integration lag” during sampling. The authors claim this deficit is asymmetric over time. It will be harmful at early timesteps by starving the model of signal needed to initiate transport, but beneficial near the end by implicitly denoising. They introduce two methods: a training-based Magnitude-Aware Flow Matching (MAFM) loss that explicitly supervises velocity magnitude early in training, and a training-free inference-time Scale Schedule Corrector (SSC) that rescales velocities with a simple time-dependent factor. Extensive experiments on ImageNet and text-to-image benchmarks suggest that SSC, despite its simplicity, can substantially improve FID and sampling efficiency without retraining.

**Compliance With Llm Reviewing Policy:**

Affirmed.

**Final Justification:**

Overall, I appreciate the paper’s insights into the “velocity deficit” phenomenon in flow matching models, as well as the proposed simple yet effective remedy. However, the current theoretical analysis relies on several non-rigorous arguments that require substantial clarification. Considering the progress in the discussion period, the authors should integrate the stated assumptions and empirical analysis more explicitly. Further narrowing the claim and scope, and making the assumptions clear upfront, will strengthen the overall contribution.

**Key Questions For Authors:**

See the weaknesses above.

**Limitations:**

No, the paper should at least discuss technical limitations, including sensitivity to hyperparameters.

**Strengths And Weaknesses:**

### Strengths

1. **Simple and practical inference-time intervention**
   - SSC is extremely lightweight, easy to implement, and does not require retraining.
   - The idea of time-dependent velocity rescaling is likely to be useful in practice, regardless of the theoretical framing.

2. **Broad empirical scope**
   - Experiments span ImageNet, high-resolution generation, and text-to-image tasks.
   - The method appears to generalize across architectures (SiT, REPA, MMDiT).

### Weaknesses

1.  **Theoretical analysis is flawed and overstated**

- **Equation (3)**
  - The inequality should be **≤**, not **<**, unless strict conditions are proven. As written, it is mathematically incorrect.
  - No conditions are given under which strict inequality holds almost surely.

- **Equation (4)**
  - The approximation relying on high-dimensional orthogonality is unjustified and informal.
  - No concentration bounds, asymptotic arguments, or error terms are provided.
  - The claim is presented as a fact rather than a heuristic.

- **Posterior expectations at boundaries (Equations 5 and 7)**
  - The assumption $\mathbb{E}[x_1 \mid x_t = x_0] \approx 0$ is generally false.
  - A simple counterexample: if the data distribution is concentrated or low-entropy, the posterior mean is non-zero.
  - The same flaw applies symmetrically to Equation (7).

- **Overall issue**
  - The analysis relies on intuition and informal reasoning rather than rigorous derivations.
  - No bounds, assumptions, or formal probabilistic statements are clearly stated.
  - Conclusions are much stronger than what the math supports.

 2. **Baseline results appear inaccurate or misleading**

The reported numbers conflict with well-known and reproducible results from SiT[1] and REPA[2]:

- **ImageNet 256×256**
  - SiT-XL/2 without CFG should reach ~8.3 FID, not 13.68.
  - REPA baselines are significantly underreported (e.g., 10.60 vs. expected ~5.9).

- **Training at 400k iterations**
  - SiT-XL/2 + REPA should reach ~7.9 FID, not 10.89.

- **Text-to-image (MS-COCO)**
  - REPA baselines are reported as 6.03 instead of ~4.14.

These discrepancies raise concerns about:
- Whether baselines were properly trained.
- Whether evaluation protocols match prior work.
- Whether comparisons are fair.

Without correcting these numbers, the claimed improvements are not credible.

3. **Hyperparameter choices are weakly justified**

- The choice of $ S_{\text{start}} = 1.1$ appears arbitrary.
- Table 2 suggests that 1.1 is not consistently optimal.
- No principled explanation, sensitivity analysis, or theoretical guidance is provided.
- The same critique applies to the MAFM weighting schedule.

Given that the method’s effectiveness hinges on these values, this is a significant omission.

### References
[1] Ma, Nanye, et al. "Sit: Exploring flow and diffusion-based generative models with scalable interpolant transformers." European Conference on Computer Vision. Cham: Springer Nature Switzerland, 2024.

[2] Yu, Sihyun, et al. "Representation alignment for generation: Training diffusion transformers is easier than you think." arXiv preprint arXiv:2410.06940 (2024).

---

> ### Author Rebuttal · Authors · 2026-03-29
>
> We thank the reviewer for acknowledging our SSC method's practicality and broad generalization. We address the detailed technical concerns below, clarifying key mathematical assumptions and experimental settings (particularly regarding the NFE budgets) that we will make more prominent in the camera-ready version.
>
> ---
> ### W1: Clarifications on Theoretical Analysis
>
> Our analysis is grounded in standard image flow matching frameworks. We will make the underlying conditions more explicit:
>
> > **Concern 1:** The inequality should be $\leq$, not $<$, unless strict conditions are proven.
>
> The strict inequality $<$ is mathematically guaranteed here because the mapping from intermediate states $x_t$ to boundary conditions is not bijective due to **trajectory crossings** in standard independent coupling. Consequently, the target velocity at $x_t$ follows a distribution $p(v_{\text{target}} \mid x_t)$ with **non-zero variance**. By applying Jensen's Inequality to the strictly convex squared norm over a variable with non-zero variance, the strict inequality holds. We will explicitly formalize the strict condition in Sec 3.1.
>
> > **Concern 2:** High-Dimensional Orthogonality approximation is unjustified and informal.
>
> We base this decomposition on the well-established geometric properties of high-dimensional spaces. As cited in Sec 3.2 (*Ledoux, The concentration of measure phenomenon.*), independently sampled high-dimensional random vectors (data $x_1$ and noise $x_0$​) are almost surely orthogonal. We will add a formal probabilistic bound for Eq. (4) in appendix.
>
> > **Concern 3:** $\mathbb{E}[x_1 \mid x_0] \approx 0$ is generally false (e.g., for low-entropy data).
>
> We completely agree that for arbitrary data, the posterior mean is non-zero. However, as explicitly stated in our Implementation Details (Sec 5.1), we strictly **followed the standard ADM preprocessing protocol, where all input data $x_1$​ are deterministically normalized to a strict zero mean**. Because $x_1$​ is strictly zero-mean and independently coupled with the noise $x_0$, $\mathbb{E}[x_1 \mid x_0] \approx 0$ holds exactly. We will add a footnote clarifying this context.
>
> ---
> ### W2: Clarifications on Baseline Results and NFE Settings
>
> We agree a fully converged SiT-XL/2 achieves ∼8.3 FID **but with NFE=250, which our reproductions perfectly match(FID=8.65, our manuscript reported in Sec 5.2.3 and Fig 3)**. The reviewer's noted discrepancy occurs because **our main tables report baselines at NFE=50 to highlight SSC's efficiency**.
>
> When strictly aligned by NFE (Tab R3):
>
> 1.**At NFE=250**: Our reproduced baselines match the original papers (e.g., SiT-XL/2 FID 8.65 vs. 8.3). Applying SSC further improves this to 6.26 (27.6%).
>
> 2.**At NFE=50 (Main Paper Setting)**: SSC drastically improves upon the lagging baselines across all models.
>
> 3.**Efficiency vs. Quality**: Most importantly, our SSC at NFE=50 (FID 7.58) surpasses the fully converged baseline at NFE=250 (FID 8.65), achieving a **5$\times$ reduction** in inference cost. (Detailed in Sec 5.2.3 and Fig 3).
>
> **Table R3: FID Comparison between Baselines and Our SSC across different NFE settings.** (* denotes reported results from original papers; others are our reproductions.)
>
> | Dataset & Setting | Method | FID (NFE = 250) | FID (NFE = 50) |
> | :--- | :--- | :--- | :--- |
> | ImageNet 256×256 (SiT-XL/2) | Baseline | *8.30 / 8.65 | 13.68 |
> | | **Our SSC** | **6.26 (↓ 27.6%)** | **7.58 (↓ 44.6%)**  |
> | ImageNet 256×256 (REPA SiT-XL/2 @ 4M) | Baseline | *5.90 / 5.97 | 10.60 |
> | | **Our SSC** | **4.58 (↓ 23.3%)** | **5.99 (↓ 43.5%)**  |
> | ImageNet 256×256 (REPA SiT-XL/2 @ 400k) | Baseline | *7.90 / 7.75 | 10.89 |
> | | **Our SSC** | **7.59 (↓ 3.9%)** | **7.69 (↓ 29.4%)**  |
> | MS-COCO (Text-to-Image) (MMDiT + REPA) | Baseline | ***4.14 / 4.4** | 6.03 |
> | | **Our SSC** | 4.47 (↑ 1.6%) | **4.71 (↓ 21.9%)**  |
> ---
> ### W3: Justification for Hyperparameter Choices
>
> We appreciate the reviewer's focus on hyperparameter sensitivity. We would like to clarify that comprehensive ablations for these choices have been fully detailed in our submission:
>
> 1.**Robustness of $s_{start}=1.1$**: As stated in line 711 (Appendix C), this was solely searched on the smallest model (SiT-S/2) and frozen. Its consistent effectiveness across massive architectural scaling and different modalities demonstrates that it corrects a universal geometric bias rather than being overfitted.
>
> 2.**Sensitivity and Schedule Shapes**: Appendix B.1 (Fig 7 and Tab 7) provide a comprehensive sensitivity analysis of different temporal injection shapes (Linear, Cosine, Quad In, Quad Out) under a strictly controlled total energy budget.
>
> 3.**Ablation of Values**: Sec 5.4 and Tab 5 explicitly ablate various scaling combinations (e.g., 1.05→1.05, 1.0→1.1, 1.1→1.0). We detail exactly why the 1.1→1.0 schedule is optimal: it provides the necessary kinetic energy to repair early structural defects while preserving the natural denoising contraction at t→1.

---

> > ### Author Rebuttal · Reviewer_YzjE · 2026-04-02
> >
> > I appreciate the reviewers’ clarification regarding the baseline results, which addresses one of my primary concerns. However, the issues in the theoretical analysis remain significant and cannot be overlooked.
> >
> > First, the strict inequalities $<$ claimed in the analysis can only hold under non-degenerate data distributions and independent data couplings. These assumptions are not clearly stated and do not generally apply in the settings considered.
> >
> > Second, the orthogonality argument relies on marginal independence between vectors. In the analysis, however, the decomposition is applied under conditioning on $x_t$, where such independence no longer holds. As a result, the cross term cannot be dismissed without a rigorous conditional concentration or decorrelation argument.
> >
> > Third, the authors conflate **global centering of the dataset** with the **conditional posterior mean**. While ADM-style preprocessing enforces $\mathbb{E}[x_1] = 0$ marginally, this does not imply $\mathbb{E}[x_1 \mid x_0] = 0$ for general data distributions. This issue is further exacerbated when the coupling is not independent—for example, under optimal transport or other structured couplings.
> >
> > Overall, I appreciate the authors’ efforts and the insights provided. Nevertheless, the current theoretical analysis contains several non-rigorous steps that require substantial clarification and strengthening before the claims can be fully justified.

---

> > > ### Author Response · Authors · 2026-04-07
> > >
> > > We thank the reviewer for the rigorous scrutiny, which is highly constructive and helps present our theory with greater mathematical rigor. We highlight two key takeaways:
> > > * **1. Our analysis is grounded in Independent Data Coupling. This is a mainstream standard**, not a restrictive edge case. It drives standard formulations and our baselines.
> > > * **2. Empirical verification closes the loop:** Measurements on high-dimensional latents validate our theoretical bounds.
> > >
> > > These clarifications strengthen, rather than alter, our original physical conclusions. We will explicitly incorporate the following formalizations into camera-ready version.
> > >
> > > **1. Scope & Foundational Assumptions**
> > >
> > > While structured couplings (e.g., mini-batch/global OT, 2-Rectified Flow) aim to restrict or eliminate trajectory crossings to minimize conditional variance, scalable continuous generative models (e.g., SiT, REPA) are practically trained using Independent Data Coupling. As Albergo et al. (*2023, Stochastic Interpolants with Data-Dependent Couplings*) note: *"Standard formulations... construct generative models built upon an independent coupling"*. Here, marginal independence holds, conditional variance remains positive due to trajectory crossings, and the MSE objective inevitably causes velocity contraction.
> > >
> > > *Action:* Section 3 will begin with:
> > > * **Assumption 1**: *The joint distribution factorizes as $q(x_0, x_1) = p_0(x_0)p_1(x_1)$, with non-degenerate marginals.*
> > >
> > > **2. Marginal vs. Conditional Independence (Eq. 4)**
> > >
> > > The reviewer correctly points out that dismissing the cross-term $\langle x_1, x_0 \rangle$ given $x_t$ requires justification. We show it vanishes at boundaries and is negligible interiorly.
> > >
> > > **A. Boundary Exactness ($t \to 0, 1$)**
> > >
> > > Under independent coupling, $x_0$ reveals no information about $x_1$. With zero-mean preprocessed data ($\mathbb{E}[x_1] = 0$):
> > > $$\lim_{t \to 0} \mathbb{E}[\langle x_1, x_0 \rangle \mid x_t] = \langle \mathbb{E}[x_1 \mid x_0], x_0 \rangle = \langle \mathbb{E}[x_1], x_0 \rangle = 0$$
> > > By symmetry, this vanishes as $t \to 1$. Thus, the deficit at critical launch/landing phases (motivating our SSC) is unaffected.
> > >
> > >
> > > **B. Asymptotic Negligibility in the Interior ($t \in (0, 1)$)**
> > >
> > > To understand the interior dynamics, assuming a standard linear interpolant path under an isotropic Gaussian proxy ($x_1 \sim \mathcal{N}(0, \sigma_1^2 I_D)$, $x_0 \sim \mathcal{N}(0, I_D)$), the conditional expectation expands to:
> > > $$\mathbb{E}\left[ \langle x_0, x_1 \rangle \mid x_t \right] = (1-t)t \frac{\sigma_1^2}{\sigma_t^2} \left( \frac{\Vert x_t\Vert^2}{\sigma_t^2} - D \right)$$
> > > By Measure Concentration, $\frac{\Vert x_t\Vert^2}{\sigma_t^2} \sim \chi^2_D$ tightly bounds the numerator to $\mathcal{O}(\sqrt{D})$. Since the principal kinetic energy (the denominator) scales linearly with dimension as $\Theta(D)$, the relative error from dropping the cross-term strictly converges to $\mathcal{O}(1/\sqrt{D})$. For $D=4096$, this error is $\approx$ 1.56%.
> > >
> > > **Empirical Verification on Real ImageNet**
> > >
> > > To verify this holds for real, non-Gaussian data without conditional estimation artifacts, we directly analyze the per-sample relative cross-term magnitude $\rho = 2|\langle x_0, x_1 \rangle| / (\Vert x_0\Vert^2 + \Vert x_1\Vert^2)$. Since $2|\langle x_0, x_1 \rangle| = \rho \cdot (\Vert x_0\Vert^2 + \Vert x_1\Vert^2)$ pointwise, taking $\mathbb{E}[\cdot \mid x_t]$ yields:
> > >
> > > $$\frac{|2\mathbb{E}[\langle x_0, x_1 \rangle \mid x_t]|}{\mathbb{E}[\Vert x_0\Vert^2 + \Vert x_1\Vert^2 \mid x_t]} \leq \frac{\mathbb{E}[2|\langle x_0, x_1 \rangle| \mid x_t]}{\mathbb{E}[\Vert x_0\Vert^2 + \Vert x_1\Vert^2 \mid x_t]} \leq \sup \rho$$
> > >
> > > Across 50K ImageNet VAE latent pairs, $\rho$ exhibits extreme measure concentration consistent with the $\mathcal{O}(1/\sqrt{D})$ high-dimensional limit:
> > > * Mean $\rho$ = 1.24%
> > > * 99th percentile = 3.98%
> > > * Absolute observed maximum = 7.27%
> > >
> > > Since $\rho$ < 4% for over 99% of the sample space and never exceeds 7.3%, the conditional relative error from omitting the cross-term is bounded at this scale—confirming the approximation is tight for complex real-world data, without relying on idealized Gaussian assumptions.
> > >
> > > *Action:* We will update Sec 3 to clarify this assumption and add a quantitative proof Appendix.
> > >
> > > **3. Conflating Global & Conditional Mean (Eqs. 5, 7)**
> > >
> > > We agree that under structured couplings ($\mathbb{E}[x_1 \mid x_0] \neq 0$), this simplification fails. However, under the practical formulations discussed in Point 1 (Independent Coupling), $x_0$ reveals no information about $x_1$, so $p(x_1 \mid x_0) = p(x_1)$. At the boundary $t \to 0$, $x_t \to x_0$. Thus, conditioning on $x_t$ converges almost surely to conditioning on $x_0$, yielding $\lim_{t \to 0} \mathbb{E}[x_1 \mid x_t] = \mathbb{E}[x_1 \mid x_0] = \mathbb{E}[x_1] = 0$. This validates our Initial Energy Injection at the exact launch phase.
> > >
> > > *Action:* Eqs. 5 and 7 will be rewritten as asymptotic boundary limits under Assumption 1.

---

### Official Review · Reviewer_waJR · 2026-03-13

**Soundness:** 3
**Presentation:** 4
**Significance:** 3
**Originality:** 3
**Overall Recommendation:** 4
**Confidence:** 4

**Summary:**

This paper identifies the Integration Lag phenomenon arising from the properties of the Mean Squared Error (MSE) objective in high-dimensional regression. Motivated by this problem, the authors propose Initial Energy Injection in two instantiations: MagnitudeAware Flow Matching (MAFM) for training and the Scale Schedule Corrector (SSC) for inference. The experimental results validate the authors' observations and demonstrate the efficiency of their methods.

**Compliance With Llm Reviewing Policy:**

Affirmed.

**Final Justification:**

My main concerns are addressed. I maintain my positive score of 4.

**Key Questions For Authors:**

1. There is a performance trade-off between SSC and MAFM (See table 1, 3). This makes it harder to assess when the training-based variant is actually preferable in practice. Can authors provide more clarification or discuss further?

**Limitations:**

The paper does not include an explicit discussion of limitations, although some trade-offs are implicitly discussed in the ablation study section.

**Strengths And Weaknesses:**

Strength:

1. The motivation is clear, and the problem is well-formulated. Figure 1 is adorable, and the writing flows smoothly.

2. The empirical observations and theoretical analysis of the Integration Lag phenomenon are solid and provide strong evidence supporting the proposed methodology.

3. The experiment results demonstrate the efficiency of the proposed method, especially the SSC training-free method.



Weakness:

1. The paper presents two methods: MAFM and SSC, in which MAFM is used for training and SSC for inference. Lack of a combined MAFM+SSC setting in experiments makes the empirical picture feel somewhat incomplete.

2. The paper considers only a linear weighting schedule (See line 262) for MAFM and does not examine whether alternative choices of $\lambda(t)$ affect performance.

---

> ### Author Rebuttal · Authors · 2026-03-29
>
> We sincerely thank Reviewer waJR for appreciating our clear motivation, smooth presentation (and glad you enjoyed Figure 1!), and for acknowledging that our theoretical and empirical analyses provide solid evidence for the Integration Lag phenomenon.
>
> ---
> ### W1 & Q1: The MAFM+SSC combination and when to prefer which?
>
> This is a highly insightful question. We have conducted the requested combination experiments across all SiT architectures. As shown in **Table R1**, there is a clear dichotomy between macroscopic global structure (measured by FID) and microscopic local texture (measured by sFID).
>
> **Table R1: Performance among SSC, MAFM, and MAFM+SSC across SiT architectures**
>
> | Model | Method | FID ($\downarrow$) | sFID ($\downarrow$) |
> | :--- | :--- | :---: | :---: |
> | **SiT-S/2** | FM Baseline | 64.83 | 16.00 |
> | | Our MAFM | 57.25 | 8.53 |
> | | Our SSC | 52.85 | 7.83 |
> | | Our MAFM+SSC ($s_{start}{=}1.1$) | **52.46** | 11.89 |
> | | Our MAFM+SSC ($s_{start}{=}1.05$) | 53.75 | **7.75** |
> | **SiT-B/2** | FM Baseline | 40.64 | 11.14 |
> | | Our MAFM | 35.83 | **6.12** |
> | | Our SSC | **32.72** | 6.46 |
> | | Our MAFM+SSC ($s_{start}{=}1.1$) | 33.36 | 11.19 |
> | | Our MAFM+SSC ($s_{start}{=}1.05$) | 33.54 | 6.43 |
> | **SiT-XL/2** | FM Baseline | 18.70 | 8.09 |
> | | Our MAFM | 16.68 | **4.67** |
> | | Our SSC | **14.26** | 5.59 |
> | | Our MAFM+SSC ($s_{start}{=}1.1$) | 16.30 | 10.62 |
> | | Our MAFM+SSC ($s_{start}{=}1.05$) | 15.66 | 5.76 |
>
> **1. The Trade-off Dynamics:**
> * **SSC dominates FID (Global Structure):** Explicitly scaling the vector field during inference physically forces the solver closer to the manifold, optimizing macroscopic geometry (e.g., reaching 14.26 FID on SiT-XL/2).
> * **MAFM dominates sFID (Local Texture):** Penalizing magnitude collapse during backpropagation structurally updates the spatial feature extractors, forcing the network to develop richer high-frequency gradients (e.g., reaching 4.67 sFID on SiT-XL/2).
> * **The Combination:** Stacking MAFM and a strong SSC ($s_{start}{=}1.1$) results in kinetic energy oversaturation, heavily degrading sFID (10.62) by overshooting delicate high-frequency manifolds. However, a moderate combination ($s_{start}{=}1.05$) successfully balances both metrics on smaller models (SiT-S/2).
>
> **2. When is the training-based MAFM preferable? (Answering Q1):**
> * **Compute-Constrained Deployment:** For already converged models, inference-based **SSC** is unequivocally optimal, offering massive structural repair with absolute zero computational overhead.
> * **High-Fidelity Textural Synthesis:** In specialized domains prioritizing local spatial consistency and crisp high-frequency textures (e.g., medical imaging or fine-grained generation), the training-based **MAFM** is essential to fundamentally update the network's spatial representations.
>
> ---
> ### W2: Alternative schedules for MAFM weighting ($\lambda$).
>
> We fully agree that exploring the shape of the magnitude penalty is important. Consistent with our findings on SSC schedule shapes (Appendix B.1), we tested alternative temporal decay profiles for the MAFM weighting schedule $\lambda(t)$ during training.
>
> We calibrated the new profiles (Cosine, Quad In, Quad Out) such that the total integral of the penalty remained equal to our default linear strategy. As shown in **Table R2** (evaluated on SiT-S/2), all profiles significantly improve over the baseline. While the "Quad In" profile offers a marginal gain in sFID, we retain the **Linear** schedule as our default recommendation because the performance differences are minimal, and the linear decay provides the simplest, most robust implementation.
>
> **Table R2: MAFM performance across different training penalty schedule shapes**
>
> | Method & Shape Type | Profile Shape | FID ($\downarrow$) | sFID ($\downarrow$) |
> | :--- | :--- | :---: | :---: |
> | **Flow Matching** | Constant | 64.83 | 16.00 |
> | **Linear MAFM** (default) | Linear Decay | 57.25 | 8.53 |
> | **Cosine MAFM** | Smooth | 57.48 | 8.89 |
> | **Quad In MAFM** | Sustain | **55.41** | **7.22** |
> | **Quad Out MAFM** | Rapid | 60.07 | 10.76 |
>
> ---
> ### Explicit Discussion of Limitations
>
> We sincerely thank all reviewers for pointing out the absence of a dedicated limitations section. We will explicitly add a "Limitations and Broader Impacts" section in the camera-ready version incorporating boundaries identified during this review:
>
> * **Dependence on High-Dimensional Orthogonality:** Our derivation relies on high-dimensional concentration of measure. Applying our energy injection to ultra-low-dimensional toy datasets may cause the solver to overshoot the data manifold.
> * **Hyperparameter Sensitivity in Domain Shift:** While our default schedules generalize robustly across natural images (ImageNet) and text-to-image tasks (MS-COCO), transferring to highly specialized domains may require re-calibrating the optimal trade-off between structural repair and textural denoising.

---

> > ### Author Rebuttal · Reviewer_waJR · 2026-04-02
> >
> > I appreciate the authors' efforts and detailed responses. My main concerns are resolved. I choose to maintain my positive score.

---

### Decision · Program_Chairs · 2026-04-30

**Decision:**

Accept (regular)

**Comment:**

The paper identifies that flow matching models systematically underestimate velocity magnitude (all reviewers), and proposes a simple, experimentally validated (all reviewers) yet heuristic (reviewers YzjE, 1W1x) method to address this problem. After the discussion, all reviewers are in favor of accepting the paper. I expect the following promises to be delivered in the camera ready version, in particular giving additional context to assumptions in the derivations (reviewer YzjE), and emphasizing that the rescaling schedule is chosen empirically (reviewers YzjE, 1W1x). Also include the additional ablations.